# ToxoNet: A high confidence map of protein-protein interactions in *Toxoplasma gondii*

Lakshmipuram S. Swapna[1☯], Grant C. Stevens[1,2☯], Aline Sardinha-Silva[3], Lucas Zhongming Hu[2], Verena Brand[1], Daniel D. Fusca[1¤], Cuihong Wan[2], Xuejian Xiong[1], Jon P. Boyle[4], Michael E. Grigg[3], Andrew Emili[2,5], John Parkinson[1,2,6]*

**1** Program in Molecular Medicine, Hospital for Sick Children, Toronto, Ontario, Canada, **2** Department of Molecular Genetics, University of Toronto, Toronto, Ontario, Canada, **3** Molecular Parasitology Section, Laboratory of Parasitic Diseases, NIAID, National Institutes of Health, Bethesda, Maryland, United States of America, **4** Department of Biological Sciences, Dietrich School of Arts and Sciences, University of Pittsburgh, Pittsburgh, Pennsylvania, United States of America, **5** Department of Biology and Biochemistry, Boston University, Boston, Massachusetts, United States of America, **6** Department of Biochemistry, University of Toronto, Toronto, Ontario, Canada

☯ These authors contributed equally to this work.
¤ Current address: Department of Ecology & Evolutionary Biology, University of Toronto, Toronto, Ontario, Canada
* john.parkinson@utoronto.ca, jparkin@sickkids.ca

**Data Availability Statement:** The raw mass spec data is available through the MassIVE database (https://massive.ucsd.edu/ProteoSAFe/static/

## Abstract

The apicomplexan intracellular parasite *Toxoplasma gondii* is a major food borne pathogen that is highly prevalent in the global population. The majority of the *T. gondii* proteome remains uncharacterized and the organization of proteins into complexes is unclear. To overcome this knowledge gap, we used a biochemical fractionation strategy to predict interactions by correlation profiling. To overcome the deficit of high-quality training data in non-model organisms, we complemented a supervised machine learning strategy, with an unsupervised approach, based on similarity network fusion. The resulting combined high confidence network, ToxoNet, comprises 2,063 interactions connecting 652 proteins. Clustering identifies 93 protein complexes. We identified clusters enriched in mitochondrial machinery that include previously uncharacterized proteins that likely represent novel adaptations to oxidative phosphorylation. Furthermore, complexes enriched in proteins localized to secretory organelles and the inner membrane complex, predict additional novel components representing novel targets for detailed functional characterization. We present ToxoNet as a publicly available resource with the expectation that it will help drive future hypotheses within the research community.

## Author summary

*Toxoplasma gondii* is a food-borne parasite that chronically infects 1 in 3 worldwide. It can cause blindness in children and is life-threatening during pregnancy and to those with compromised immune systems. During infection, *Toxoplasma* relies on the coordination of suites of proteins that allow it to invade host cells and persist. In this study we combined a sophisticated biochemical separation platform with a novel machine learning

massive.jsp) with the accession number MSV000092051. Other datasets used in this article are included as supplemental tables. Raw peptide counts associated with each fraction are available without restriction at https://github.com/ParkinsonLab/ToxoNet.

**Funding:** This research was supported by a grant to JP, AE and JB from the National Institutes of Health (R21AI126110), to JP, AE and MEG from the Canadian Institutes for Health Research (PJT 152921), and in part by the Intramural Research Program of the National Institute of Allergy and Infectious Diseases (NIAID) at the National Institutes of Health to MEG. GCS was supported by a student Restracomp fellowship given by the Hospital for Sick Children. The funders had no role in study design, data collection and analysis, decision to publish, or preparation of the manuscript. MEG receives a salary from the Intramural Research Program of the National Institute of Allergy and Infectious Diseases (NIAID) at the National Institutes of Health.

**Competing interests:** The authors have declared that no competing interests exist.

strategy to build a global roadmap that describes how parasite proteins work together as molecular machines to help drive infection. Our roadmap, which we term ToxoNet, details 2,063 connections between 652 proteins. These include proteins representing parasite-specific innovations that help *Toxoplasma* perform essential functions such as the generation of energy. Further, in addition to capturing proteins already implicated in host cell invasion pathways, we identified many additional components of these pathways that represent novel targets for future drug development strategies.

## Introduction

*Toxoplasma gondii*, the causative agent of toxoplasmosis, is an apicomplexan intracellular parasite of biomedical importance, estimated to infect 30% of the world's population [1]. It is the leading cause of infectious retinitis in children and is life-threatening in pregnancy and to the immunocompromised [2–4]. Despite its impact, viable vaccines have yet to be developed and few treatments are available. Further, resistance to front line drugs (sulfonamides) is emerging [5]. Able to form tissue-cysts, *Toxoplasma* exhibits a heteroxenous lifestyle, with a sexual phase occurring in the intestinal epithelium of cats and an asexual phase capable of infecting any nucleated cell of any warm-blooded animal. To fulfil its life cycle, *Toxoplasma* is exquisitely adapted to exploit its hosts. For example, to invade its hosts requires specialized processes that mediate host cell attachment, penetration and modulation of host pathways that prevent parasite clearance. Driving this process are hundreds of invasion-related proteins and complexes, that directly impact virulence [6–13]. Key systems include the inner membrane complex (IMC) which drives motility and cell division [14], SAG-1 related sequence (SRS) proteins which are parasite surface receptors involved in host cell recognition and attachment [13,15], microneme proteins, that are also involved in gliding motility and host cell attachment [16], and dense granule and rhoptry proteins which are secreted during and after host cell invasion to both help the parasite enter the host cell and modulate host cell behaviour subsequently [17–21]. Genome analyses of *Toxoplasma* strains, exhibiting different virulence phenotypes, reveal many of the genes encoding these proteins exhibit significant genetic variation [9]. To better understand the involvement of these genes in pathways driving host cell invasion, a number of approaches such as protein microarrays [22], CRISPR screens [23], phosphoproteome analysis [24,25], coexpression analysis [6] and metabolic modeling [26] have been applied to reveal the contribution of many of these proteins to pathogenesis. However, information on how these proteins are organized into physical protein complexes is lacking. Such an understanding is important as it allows functions to be ascribed to otherwise uncharacterized proteins through *guilt by association* [27], as well as providing mechanistic insights into how proteins are coordinated to perform specialized biological processes.

Over the past two decades, a number of methods have been developed to help elucidate the physical protein-protein interactions that define protein complexes, including yeast two-hybrid (Y2H) screens [28], affinity purification mass spectrometry (AP-MS) [29], spatially restricted enzyme labelling techniques (e.g. BioID [30] and APEX [31]), phage display [32] and protein microarrays [33]. For example, in high throughput applications, AP-MS has been applied to generate large scale maps of protein-protein interactions for both yeast and *E. coli*, each comprising hundreds of protein complexes [34,35]. However such approaches are labor intensive and expensive, requiring the generation of thousands of cell lines carrying tagged gene constructs. Instead, biochemical cofractionation (or coelution) [36,37], has emerged as an efficient and effective route for elucidating protein complexes in a near native

pathophysiological context. In a typical application, native protein lysates are separated into hundreds of biochemical fractions, each subjected to shotgun proteomics. Coeluting proteins identified in the same fractions are then considered to form stable interactions. Applying this approach on a global-scale has resulted in the recovery of hundreds of stably associated soluble protein complexes for humans [36] and metazoans [37]. More recently, this strategy has been applied to generate smaller sets of complexes for *Trypanosoma brucei* [38,39] and *Plasmodium* [40]. Since coelution can result in protein pairs that do not physically associate (false positive interactions), rigorous computational scoring procedures that integrate additional supporting functional association evidence (e.g. based on expression profiles or literature evidence) are required to increase the overall quality of the predicted interactions.

In this study, we applied a stringent coelution strategy to generate the first genome-scale physical protein interaction network for *T. gondii*. Based on biochemical coelution data, we apply a novel computational strategy, to integrate additional functional genomics data, including orthogonal coexpression, phylogenetic profiles and known domain-domain interactions. Since non-model organisms often lack the depth of known protein complexes of model organisms, required for training supervised machine learning approaches, we complement an approach based on a Random Forest classifier, with an unsupervised approach based on Similarity Network Fusion (SNF) [41], which provides superior performance than random in the absence of suitable training data. Together this pipeline yielded a combined network of 3,753 interactions capturing 792 proteins which was further filtered to obtain a combined high confidence network of 2063 interactions connecting 652 proteins. We partitioned this network to define distinct protein complexes including assemblies that have previously been characterized, as well as novel macromolecules involving invasion proteins that represent novel targets for detailed functional characterization.

## Results

### Coelution profiling coupled with supervised machine learning recapitulates known protein complexes

This study is the first effort to generate a genome-wide protein-protein interaction network for the apicomplexan parasite *Toxoplasma gondii*, using a biochemical coelution approach, augmented by the integration of functional genomics datasets. To generate coelution profiles for *T. gondii*, ME49 tachyzoites were harvested from human foreskin fibroblasts, and subject to six biochemical fractionation experiments: five using beads featuring different surface chemistries for selective enrichment of different proteins followed by standard HPLC separation [42], and one using high performance mixed-bed IEX (**Fig 1A and S1 Table**), resulting in in the collection of 60 fractions from each of the five beads and 120 fractions from IEX for a total of 420 fractions. Consistent with previous co-elution studies involving parasites [38,39], by focusing on a diverse fractionation strategy involving six complementary experiments, we aimed to maximize the number of proteins recovered. We appreciate that the six experiments do not represent true biological replicates, however previous co-elution studies have shown replicates to be highly reproducible [43]. Furthermore through using the same lysate for each bead elution, our strategy offers the potential to capture the same protein interactions across multiple experiments. The proteins in each fraction are proteolytically cleaved into peptides, which were subsequently identified by precision mass spectrometry (LC-MS) at an estimated false discovery rate of 5%, yielding a total of 1,423 unique *T. gondii* proteins or which 187 were common to all six fractionations (**S2 Table**). One-half of these proteins are annotated with Gene Ontology (G.O.) terms [44], while another one-third are defined as 'hypothetical proteins' (**Fig 1B**). Notably, we identified 105 invasion related proteins, including 17 IMC

**Fig 1. Network generation, statistics and overview analyses.** (A) An outline of the methodology employed in this study for generating the predicted *T. gondii* network. In six separate elution experiments, involving five bead purifications and one from whole cell lysate, we generated a total of 420 fractions. Utilizing six complementary scoring schemes (see Materials and Methods), a total of 45 coelution similarity scores were calculated. In a supervised machine learning approach, of the 45 coelution similarity scores, 28 were deemed informative for predicting interactions (**S3 Table**) and

integrated with five functional genomics scores to generate a *supervised* network. In a separate unsupervised approach, we combined three coelution similarity scores, with three functional genomics scores to generate an *unsupervised* (fused) network. (B) Functional characterisation of the proteins identified by coelution in this study (with ≥ 5 spectral counts), with special reference to invasion clusters. (C) Distribution of protein abundance for cofractionated proteins. The red dashed line indicates the cutoff for filtering proteins with low spectral counts. (D) Receiver operating characteristic (ROC) curve for supervised machine learning using RandomForest. The RandomForest output corresponding to a false positive rate of ~0 and true positive rate of ~0.3 on the graph (indicated using a *—value of 0.57) was used as the cutoff for selecting the high confidence network. (E) Network statistics of the combined network and combined high confidence network, with a breakup into supervised and unsupervised networks for the full network. (F) The predicted protein-protein interaction network of *T. gondii*.

proteins, 9 rhoptry neck proteins (RONs), 17 microneme proteins (MICs), 27 rhoptry proteins and 35 dense granule proteins (GRAs). Although ~400 proteins were identified at relatively low abundances (in terms of total spectral counts), ~1000 proteins were robustly identified with ≥5 spectral counts, implying substantive abundance (**Fig 1C**). To ensure we include only proteins with reliable identifications in our analyses, during generation of the protein-protein interaction networks, as detailed below, we removed proteins with less than five spectral counts.

To define interactions between these proteins, the coelution profiles were used to generate six complementary coelution similarity scores capturing each possible coeluting protein pair per experiment: Pearson correlation coefficient with noise modeling (PCCNM); weighted cross-correlation (WCC); co-apex; mutual information; topological overlap similarity based on PCCNM; and topological overlap similarity based on WCC (**S3 Table**)(see Methods for detailed description). These scoring schemes were calculated individually for the 6 experiments and then once more for the elution profiles combining all experiments, which generated a more informative profile for proteins coeluting consistently in multiple experiments. In all, a set of 45 coelution similarity scores were used to define overlaps in coelution profiles. Given that certain proteins can elute together by chance, rather than due to physical association, we next integrated additional functional genomics datasets as a filter to define functional genomics scores. These included two scoring schemes based on gene-coexpression datasets [6]; two scoring schemes based on phylogenetic profile datasets [45]; a single scoring scheme based on domain-domain interactions [46]; and two scoring schemes based on the STRING database [47] within a supervised machine learning framework (RandomForest–see Methods). For training purposes, we used a gold standard set of positive interactions curated from orthologues of known protein complexes collected from the CORUM [48] and Cyc2008 [49] databases, together with interactions inferred from the Toxocyc resource [50] and G.O annotations [51] (**S4 Table**). Gold standard negative interactions were generated based on differences in cellular localization (see Methods). Out of a total of 52 scores, feature selection identified 33 to be informative (**S3 Table**). Of these, 28 correspond to coelution similarity scores and 5 to functional genomics scores. Notably, scores obtained from the STRING database were not informative, likely due to their relatively low representation in the dataset, and were therefore not considered further. Based on the 33 of the 52 scores identified by feature selection, we used 10-fold cross validation to show integration of these scores with functional genomics profiles using the RandomForest classifier yielded the best performance (AUC = 0.806; **Fig 1D**). This compares to the best performing individual dataset (coelution scores only; AUC = 0.722) or the combination of all functional genomics datasets (AUC of 0.749). Furthermore, precision-recall curves reveal that the combination of coelution and functional genomics datasets contributes to the highest recall and precision in contrast to using either coelution data alone or only the functional genomics data (**S1 Fig**). Using this RandomForest classifier, a test set of 174455 protein pairs with biochemical support

(corresponding to a coelution similarity score $\geq 0.5$ in at least one of the experiments) were evaluated. The resulting predictions were further pruned to remove: i) pairs involving proteins with a spectral count of $< 5$ (to remove spurious correlations arising out of low abundance) and ii) pairs involving proteins which do not contain any unique peptides distinguishing between the pair.

Based on the set of pairs predicted to be interacting according to the RandomForest analysis (corresponding to a Random Forest score $\geq 0.5$), our approach yielded a network comprising 2,833 interactions between 730 proteins. From these, we applied a RandomForest prediction score cutoff of 0.57 (corresponding to the lowest false positive rate (~0) on the ROC plot, as indicated by a * in Fig 1D) to further define a higher quality dataset comprising 1,541 interactions between 549 proteins (S2 Fig and S5 Table). It should be noted that although the FPR of 0 associated with the selected cutoff appears to be highly stringent, this rate only applies to the training set and does not preclude the presence of false positives in our predictions. This network segregates proteins associated with invasion from known complexes and functional interactions conserved across organisms. Moreover, as expected, the supervised network recapitulates many known complexes including the proteasome, the ribosome, and a snRNP complex, demonstrating its ability to capture features in the training datasets (S2 Fig). It also recapitulates previously reported interactions in *T. gondii*, including MIC1-MIC4-MIC6 [52], ROP5-ROP18 [53], GRA2-GRA4-GRA6 [54].

## Applying an unsupervised machine learning approach identifies additional protein-protein interactions not captured by the supervised approach

A major challenge for inferring protein interactions for non-model organisms is the lack of comprehensive datasets of previously characterized complexes that can serve as training data for more sophisticated machine learning algorithms. To overcome this challenge, we explored an unsupervised machine learning approach, termed Similarity Network Fusion (SNF) [41], to predict protein interactions in the absence of training data (Fig 1A). In this approach, the same scores as used for the supervised approach are used to construct 7 individual networks, based on combinations of three scoring schemes for coelution data (PCCNM, WCC, Coapex1), two scoring schemes for coexpression datasets (COEXPR-RS, COEXPR-MA), and two scoring schemes for phylogenetic profiles (MI-Pij, MI-PresAbs) (see Methods for more details). Networks are then fused using a method based on message-passing theory to identify interactions supported by multiple datatypes, eliminating poorly supported interactions and strengthening interactions supported by multiple datatypes.

We exhaustively explored all possible combinations of datasets and hyperparameters for the generation of networks using SNF (see Methods). From these networks, we focused on the network that captured the greatest number of likely complexes (corresponding to the highest number of clusters with overlap score $\geq 0.25$ [55], resulting in 943 interactions between 343 proteins (S6 Table and S3 Fig). This network was constructed using a combination of scores fused by SNF (PCCNM, WCC, Coapex1, Coexpr-RNAseq, Coexpr-Microarray, MI-pij). To further reduce the number of false positives in this dataset, we additionally filtered out interactions with coexpression scores $< 0.5$ (S4 Fig), which corresponds to the cutoff where the supervised network starts losing bone fide interactions. Consistent with the supervised network, we further removed proteins represented by fewer than 5 spectral counts, as well as pairs in which each protein contains no unique peptide capable of distinguishing between the pair. This filtered network (defined as the *unsupervised network*) consists of 523 interactions between 282 proteins. Within this network, we identified 73 hypothetical proteins and 32 invasion-related proteins, with some connected interactions localized to the same compartment (e.g., MIC4,

chitinase-like protein CLP1, TGME49_200270 are localized to the microneme, according to GO evidence codes).

Comparison of the overall unfiltered supervised and unsupervised networks reveals 62 proteins exclusive to the unsupervised network (S5A Fig). These proteins are involved in 262 interactions, including some paralogs such as HMG_box_containing_protein, aminopeptidases, MIC17A-MIC17B, MIC17A-PAN/Apple domain-containing protein, which are likely to be involved in functional interactions. The distribution of their spectral counts is similar to that of all proteins in the coelution dataset (S5B Fig). The predicted unsupervised pairwise interactions score significantly better than randomly generated interactions in terms of their highest coelution scores (PCCNM, WCC and Coapex1). However, they are consistently lower than the highest PCCNM and WCC scores for the predictions from the supervised network, but comparable in terms of the Coapex1 score. Of the remaining proteins, 415 are exclusively predicted in the supervised network whereas 315 are common to both (S5A Fig). For these common proteins, the two approaches predict 73 common interactions, the rest are mutually exclusive–with 2053 interactions predicted by the supervised approach, and 898 interactions predicted by the unsupervised approach. Based on the seven scoring schemes outlined above (i.e. PCCNM, WCC, Coapex1, COEXPR_MA, COEXPR_RS, MI_PreAbs and MI_pij), we find that the supervised approach captures interactions with higher coelution similarity scores in general (except for Coapex1 score–which is similar for both supervised and unsupervised approaches) and closer phylogenetic profiles (MI-pij score) than the unsupervised approach, which still demonstrates improved performance over sets of randomly generated interactions (S5C Fig). Combining the supervised and unsupervised approaches generates a single *combined* network of 792 proteins and 3,753 interactions (Fig 1F). The networks share 315 proteins (an overlap of 43.1% for supervised; 83.1% for unsupervised) and 73 interactions (an overlap of 4.4% for supervised; 8.2% for unsupervised). Among these 792 proteins are 385 annotated as 'hypothetical' and 105 predicted to be involved in invasion (S5 and S6 Tables).

From this combined network, a high confidence network of 549 proteins and 1541 interactions was derived for the supervised network, based on the score at a false positive rate cutoff of ~0 on the ROC curve (i.e. 0.57), and a high confidence network of 282 proteins and 523 interactions was derived for the unsupervised network, based on a coexpression score of $\geq 0.5$ (S7 Table). The combined high confidence version of ToxoNet comprises of predicted interactions from both supervised and unsupervised approaches, resulting in a network of 652 proteins and 2063 interactions, supported by biochemical coelution (PCCNM (or) WCC scores $\geq 0.5$) in at least one of the six experiments, providing a resource of known/putative interactions for several proteins involved in core cellular processes and apicomplexan-specific processes, including 68 invasion-related proteins and 130 hypothetical proteins.

## Integrative analyses validate the biological relevance of ToxoNet

To assess the quality of the combined high confidence network and its ability to recapitulate biologically meaningful relationships, we performed a series of meta-analyses. Focusing on network statistics, we find that ToxoNet exhibits properties consistent with previously published PPI networks. Notably, it exhibits small world properties including a scale-free architecture [56] and relatively short path length distributions (Fig 2A and 2B). Next, we examined the role of essential and conserved proteins in the combined high confidence network. As expected, we find that essential proteins, previously defined through a genome-wide CRISPR screen of the tachyzoite lifecycle [23], are more highly connected with significantly higher degree (p = 2.9E-10; Fig 2C). Dispensable proteins however, mediate more central roles in the network with higher betweenness (p = 1.5E-2) than essential proteins (Fig 2D). These data

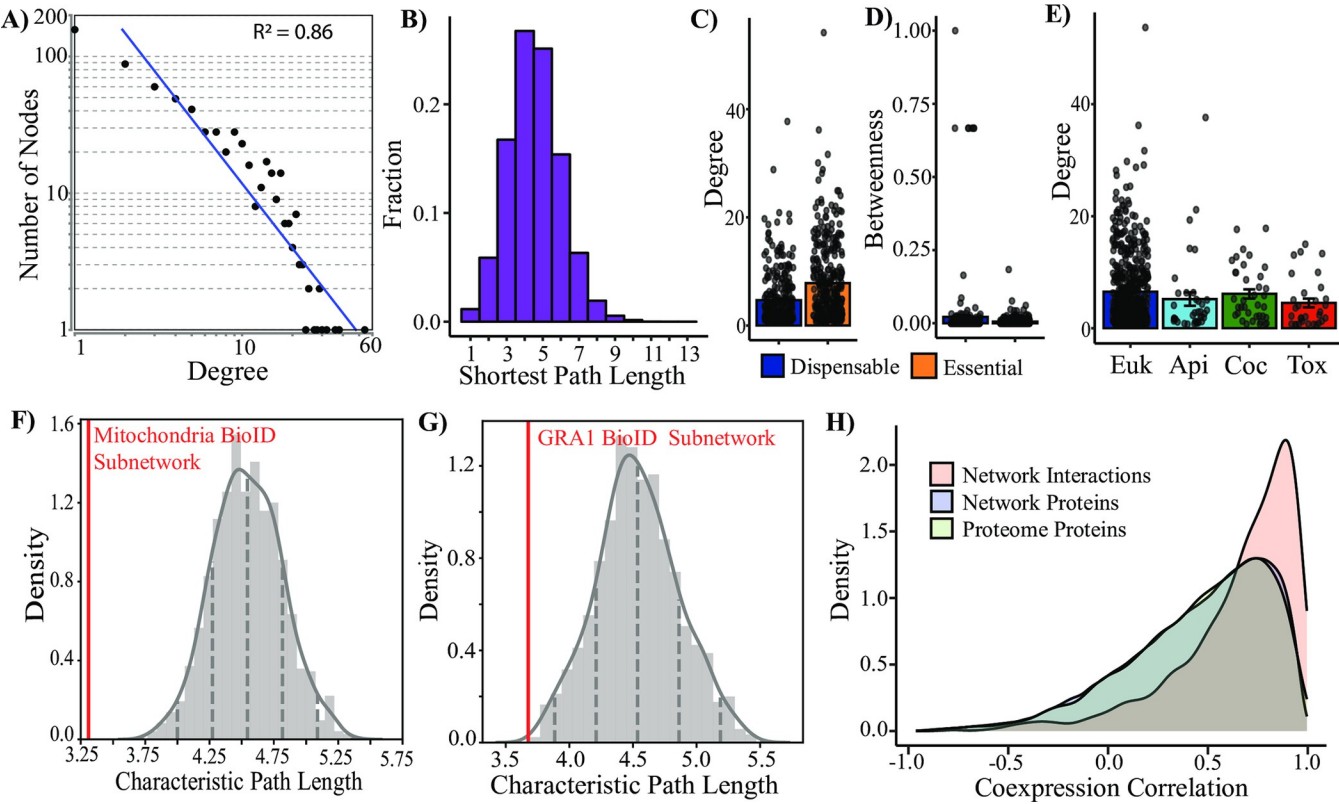

**Fig 2. Benchmarking of ToxoNet using previously published datasets.** (A) The node degree distribution with a power fit line ($R^2 = 0.86$). (B) Distribution of shortest path lengths. (C-D) The bar graphs with essential proteins (orange) and dispensable proteins (blue) indicate significant differences in node degree and betweenness centrality (p = 2.9E-10, p = 1.5E-2 respectively). (E) The bar graph indicates differences in node degree at various evolutionary timepoints. Error bars indicate standard error. (F-G) The distribution of random permutations (n = 1000) of characteristic path lengths relative to the actual characteristic path length (red line) of a set of candidates identified in BioID experiments as putative mitochondrial (n = 36, F) and dense granule (n = 23,G) proteins (p = 2.8E-6, p = 4.3E-3, respectively). (H) The distribution of coexpression in RNA-seq experiments of interacting network proteins, non-interacting network proteins and a random sampling of the proteome (n = 10,000; Kolmogorov-Smirnov p < 2.2E-16).

support the theory that essential proteins serve as network hubs with key organizational roles in PPI networks [57,58]. Likewise, we find that highly conserved proteins (with orthologs predicted in other, non-apicomplexan, eukaryotes) are also more highly connected than lineage specific proteins, albeit with significant differences only observed with *Toxoplasma*-specific proteins (p = 0.02; **Fig 2E**). These findings are consistent with previous studies from many biological networks including *E. coli* and yeast [34,59].

We further validated the quality of ToxoNet through comparisons with existing protein interaction data for *T. gondii* as well as other parasites. Recently, complementary APEX and BioID approaches [60] were applied to identify a shared set of 161 *T. gondii* proteins, predicted to localize to the mitochondria. Since we expect colocalized proteins to appear closer in our combined high confidence network, we were reassured by a significantly shorter average pathlength between the 36 proteins that were present in ToxoNet (of the 161 proteins identified in the previous study) than expected by random (average shortest path length = 3.0; p = 2.8E-6; **Fig 2F**). A similar dual screen of dense granule proteins [61] identified a common set of 33 related GRA proteins. Strikingly, the average shortest pathlength of the 23 proteins that were found in ToxoNet exhibited a relatively shorter average path length than expected (average shortest pathlength = 3.68; p = 4.3E-3; **Fig 2G**). Further datasets based only on a BioID approach [62,63], also generally exhibited smaller (not statistically significant trend)

characteristic pathlengths, with a screen of the IMC-related protein, ISP3, being notable for exhibiting a higher average pathlength than random (**S6 Fig**). This likely reflects the higher false-positive rates associated with these screens in the absence of additional experimental support. The congruency with high confidence datasets demonstrates that the spatial proximity of proteins within the cell is echoed by shorter distance within ToxoNet.

Beyond proximity screens, we also compared ToxoNet with two recently published protein interaction networks for *Trypanosoma brucei* [64] and *Plasmodium* parasites [40]. Notably both networks were generated using a similar coelution methodology. Typically, protein interaction networks often display little overlap between species [65]. However, here we found that ToxoNet exhibited a highly significant overlap in protein interactions with both *T. brucei* (283 common interactions, p = 6.4E-204) and *Plasmodium* (450 common interactions, p = 0; **S6B Fig**). While most of these shared interactions occur between highly conserved proteins, and may reflect their inclusion in training data (**S4 Table**), we did identify interactions between an apicomplexan-specific protein (TGME49_268830), with an ATP synthase subunit (TGME49_226000) as well as cytochrome c1 (TGME49_246540), to be conserved in the *Plasmodium* network. The former interaction between TGME49_268830 and TGME49_226000 has been validated in *T. gondii.* Its presence in the *Plasmodium* network supports the conservation of this interaction as an apicomplexan-specific adaptation to ATP synthase.

Next, we analyzed our combined high confidence network in the context of expression data from RNA-seq datasets that had previously been withheld from the machine learning analyses applied to generate ToxoNet. Pearson correlation values were calculated for transcript pairs across ten tachyzoite time-points from three independent experiments [11,66]. Expression of putatively interacting proteins has a significantly greater density at higher coexpression correlation values than both non-interacting network proteins and random sampling of proteome pair-wise combinations (p < 2.2E-16), whereas, the density of Pearson correlation values between non-interacting network proteins and random proteome pair-wise combinations is not significantly different (p = 0.61; **Fig 2H**). G.O. annotations of putative interactions were also compared using terms annotated to *T. gondii* proteins available on ToxoDB. Despite the limited number of annotated proteins, of those pairs in which both proteins are annotated, 54% have the same G.O. process term (p = 0), 60% have the same G.O. component term (p = 8.27E-272) and 49% have the same G.O. function term (p = 3.82E-272). Since G.O. component terms were also utilized in the construction of negative training data utilized in the supervised learning step that might enhance network performance, we also compared our predictions with a subcellular atlas of the *Toxoplasma* protein predicted using hyperLOPIT localization patterns [67] which was not included in any aspect of network generation. From this independent analysis we found a significant number (37%, p < 1.12E-203) of interacting pairs were predicted to share the same compartment (**S8 Table**). It is important to note that the hyperLOPIT dataset predicts localization and not interactions *per se.* Thus it is possible that our dataset may contain interactions between proteins that, while occupying different locations, nevertheless may share a point of contact (e.g. 40S ribosome and 60S ribosome, cytosol and mitochondrion membrane). Furthermore, in this study, we used protein fractions that were collected after solubilization of Toxoplasma cells, hence potential barriers preventing interactions between proteins were removed. These include those associated with rhoptries, micronemes and dense granules which are secreted during host invasion and have been shown to form physiologically relevant complexes (e.g. the microneme and rhoptry proteins, AMA1 and RON2 [68]). We therefore looked at interactions between proteins predicted to occupy different compartments and found an additional 28% of interacting pairs that either share a point of contact (e.g. 40S ribosome and 60S ribosome, cytosol and mitochondrion membrane) or have the potential to interact after secretion (e.g. dense granules and rhoptries, micronemes

and rhoptries). The remaining (34%) pairs represent potential interactions that can be explored in future experiments. For example microneme proteins have been shown via co-IP to interact with the cytoplasmic protein, fructose-1,6-bisphosphate aldolase [69]. Together, these results illustrate that the relative performance of ToxoNet is consistent with that found for other protein-protein interaction studies.

## ToxoNet recapitulates known protein complexes and identifies novel components and novel complexes

Since coelution data is enriched for proteins that physically interact, we applied the Cluster-ONE algorithm [55] to identify protein complexes based on their interactions (see Methods). In total we identified 93 overlapping clusters (representing putative protein complexes) with an average of 8.2 proteins per cluster (**Fig 3** and **S9 Table**). In total, 28 clusters recapitulate 25 training complexes with an overlap score of 0.25 or greater [70]. These include the protein kinase CK2 complex (cluster 72, overlap = 1), the U4/U6.U5 tri-snRNP complex (cluster 33, overlap = 0.79), the C complex spliceosome (cluster 56, overlap = 0.75), the reductive TCA cycle (cluster 54, overlap = 0.75), eIF3 complex (cluster 44, overlap = 0.51), the vacuolar ATP synthase complex (cluster 53, overlap = 0.56) and the box C/D snoRNP complex (cluster 70, overlap = 0.6). Additionally, the ClusterONE algorithm recapitulated experimentally validated complexes absent from training data, including MIC1/4/6 (cluster 83, overlap = 0.34). At a lower cut-off, we also recapitulate GRA2/4/6[11] (cluster 26, overlap = 0.23), the *T. gondii* ATP Synthase complex with Apicomplexan-specific subunits [71] (overlapping clusters 12 and 29, overlap = 0.13) and the Moving Junction (cluster 88, overlap = 0.13). The glideosome is also partially recapitulated (cluster 52, overlap = 0.11) with GAP45 and MLC1 included in the same cluster as the α,β-tubulin complex.

While we found clusters enriched in lineage-specific proteins generally localize to lineage-specific organelles (i.e. rhoptries, micronemes and dense granules), there are also examples of lineage-specific proteins in clusters with proteins that have been shown to be associated with either the cytosol or the mitochondria. Again, utilizing expression data withheld from training, we found that of the 93 clusters, 45 (48%) are predicted to be significantly coexpressed (**Fig 3B**). This is consistent with previous studies of protein complexes that display a significant enrichment of component coexpression and likely reflect regulation through common transcription factors [72]. Analysis of essential protein distributions also revealed a non-random pattern of organization (p = 1.079E-06; **Fig 3C**). Specifically, clusters defined by ToxoNet contain a higher proportion of dispensable proteins. Of the 30 clusters enriched in essential proteins ($\geq$ 75% of components are essential), 17 represent recapitulated training complexes. Interestingly, dispensable clusters, composed of $\leq$ 25% essential proteins (n = 25), are significantly enriched in proteins associated with network invasion (i.e., SRS, MIC, ROP, RON and GRA proteins) and IMC proteins (p = 7.3E-35), as well as proteins restricted to Apicomplexa (p = 1.6E-17; **S7 Fig**). In addition to the high percentage of invasion and IMC proteins per dispensable cluster (41%), these dispensable clusters are composed of an average of 25% hypothetical proteins, offering an exciting wealth of candidates for novel virulence factors. These results are again consistent with previous studies that have shown highly conserved protein complexes to be essential, while those composed of lineage-specific components, reflecting in this case, more recent adaptations to parasitism, tend to be more dispensable.

To further explore the potential impact of these macromolecular adaptations at the strain level, we examined genetic variation data associated with 80 strains of *T. gondii*. Among the interacting proteins identified in our combined high confidence network, we identified 2893 non-synonymous single nucleotide polymorphisms (SNPs) present in at least 4 strains [9,73].

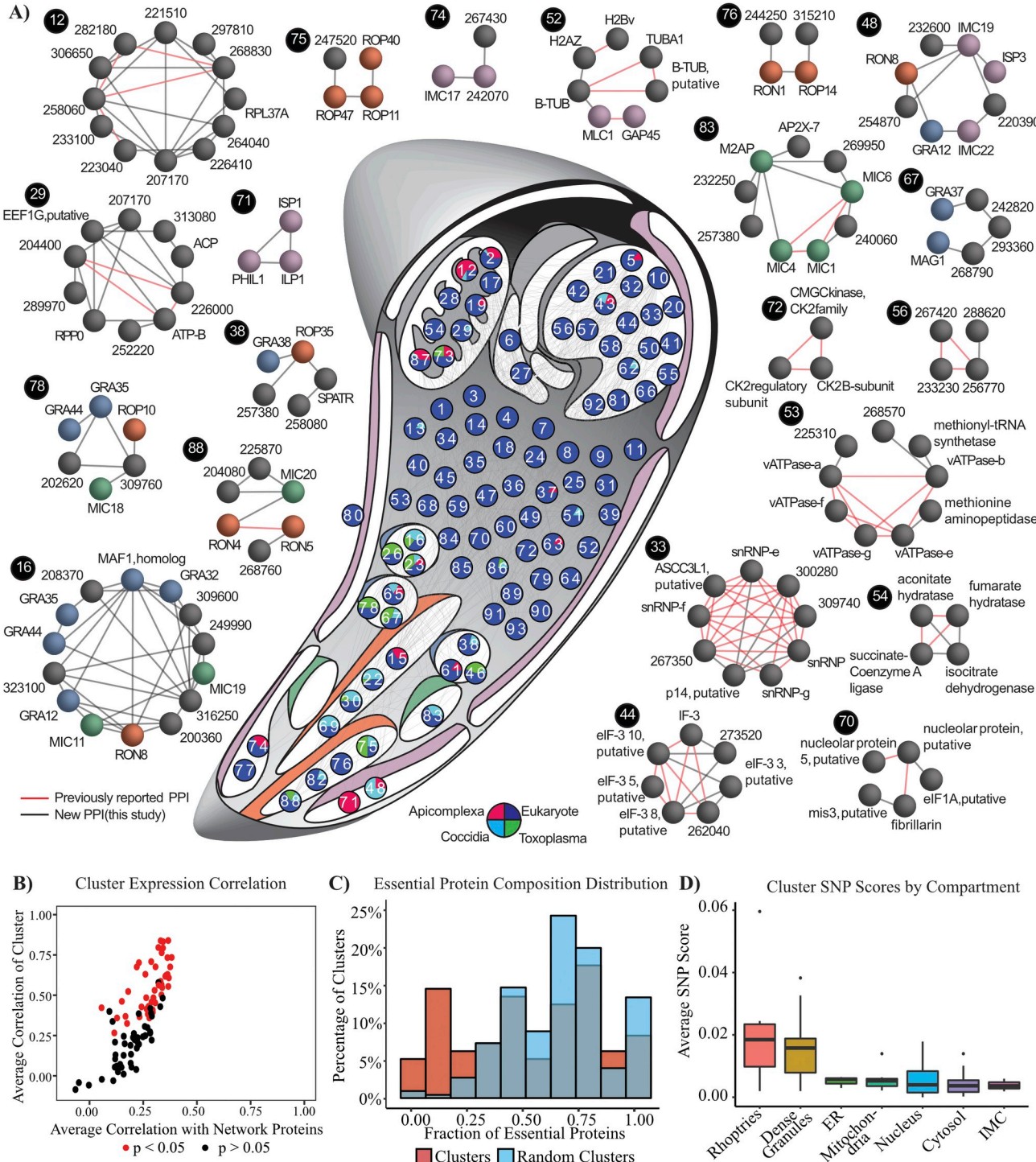

**Fig 3. Organization of ToxoNet into discrete protein complexes** (A) Global organization of predicted complexes is visualized by a graph where nodes represent clusters that are placed in their putative compartment (IMC = purple, micronemes = green; rhoptry = orange, dense granules = blue). Nodes are placed on the *T. gondii* schematic in their putative organelle based on the most frequent hyperLOPIT localization prediction. Unclassified clusters are placed in the cytosol. Numbered nodes are colored according to lineage composition (see inset). The edges indicate inter-cluster interactions. Noteworthy clusters are highlighted to the sides with known (red) and novel (black) interactions indicated between proteins. (B) Coexpression of cluster components–clusters are coloured by significance of coexpression. (C) Distribution of essential proteins in clusters and random generated clusters (n = 100). (D) Boxplots compare the average SNP scores for putative complexes across different compartments.

A SNP score was assigned to each protein based on the number of non-synonymous SNPs normalized by protein length and the average was taken across all cluster proteins to identify putative complexes with the greatest genetic variation (**S8 Fig**). Consistent with previously reported higher rates of SNPs within invasion family proteins, particularly in *GRA* and *ROP* family proteins [9], we found that putative complexes from dense granule and rhoptry compartments exhibited the highest average SNP score (**Fig 3D**). To predict which SNPs are most likely to interfere with physical interactions, we examined the Pfam domains [74] and identified premature stop codons present in the proteins of the ten complexes with the highest average SNP score. In the experimentally validated MIC1/4/6 complex (cluster 83), we identified five non-synonymous SNPs in three domains mediating interactions with other proteins (S55G and Q82E in the first TSR1 domain of MIC1, K114N and R123L in the EGF2 domain of MIC6 and A171V in the second apple domain of MIC4) [75,76]. Notably, strain members of clade E (which includes TgH21016, CASTELLS, TgH26044 and TgH20005) were unique insofar that they carry four of these five mutations, suggesting that the MIC1/4/6 complex in these strains may not occur in its canonical form. Similarly, we found that for 5 strains, including ME49, 3675, B73, PRU, and TgGoatUs21, GRA44 (TGME49_228170) orthologs contain a premature stop codon that results in a 105 amino acid truncation. Given its interactions with other proteins in three predicted invasion-related clusters (16, 23 and 78), we predict these complexes may again exhibit different patterns of organization outside these strains.

## Complex predictions identify novel apicomplexan-specific mitochondrial adaptations

Recent studies have elucidated apicomplexan-specific subunits to the protein complexes involved in the electron transport chain (ETC) and oxidative phosphorylation. ToxoNet recapitulates these features. The conserved eukaryotic α-, β-, δ-subunits and the apicomplexan-specific subunits of the ATP synthase complex (i.e., TGME49_258060, TGME49_268830, TGME49_282180 and TGME49_223040) [71,77,78] were represented in overlapping clusters 29 and 12, respectively. Furthermore, the apicomplexan adaptations to the cytochrome c oxidase (COX) complex, or ETC Complex IV, TGME49_264040, TGME49_221510 and TGME49_297810 [60], were also captured in cluster 12. Recently, TGME49_207170 was identified as a novel component of ETC Complex III of the electron transport chain (ETC) [79,80]. In general, the proteins captured in cluster 12 are enriched in predicted mitochondrial targeting sequences and an essential phenotype (**Fig 4A**). The remaining protein which remains uncharacterized and fits this profile is TGME49_306650. Its presence in this cluster predicts a role in ETC. This prediction was also supported by a parallel study which predicted TGME49_306650 to be a member of ETC Complex II [80].

To assess the localization of TGME49_306650, we engineered a *T. gondii* PRU strain with a 3×HA tag at the C-terminus of the endogenous locus. Mitochondrial localization was validated by colocalization with the mitochondrial marker, F1B ATPase (**Fig 4B**). To further characterize its binding partners, we performed immunoprecipitation (IP) of the HA tagged hypothetical protein (TGME49_306650) utilizing the same sonication-based lysis protocol which generated ToxoNet. The AP-MS data from two replicates of *T. gondii* PRU 306650-HA and *T. gondii* PRU wild-type (WT) strains were filtered utilizing SAINTexpress [81] to identify interactors. Though highly enriched for the bait, none of the interactions or cluster proteins were represented in the prey proteins at a FDR cut-off of 0.01 or less; however, some of the components displayed in Cluster 12 are present at higher FDR values, including TGME49_264040 (FDR = 0.12), TGME49_221510 (FDR = 0.17), TGME49_207170 (FDR = 0.17), TGME49_282180 (FDR = 0.17), TGME49_223040 (FDR = 0.17) and TGME49_233100

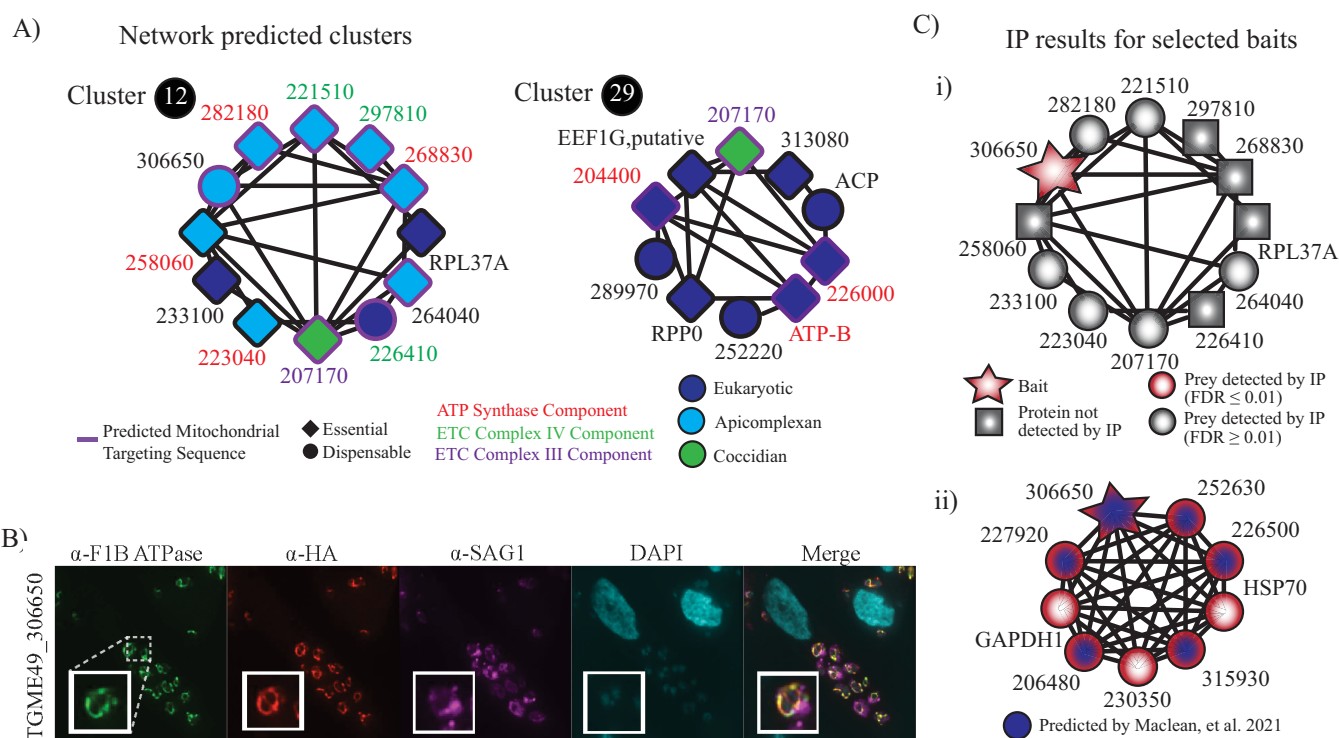

**Fig 4.** (A) Clusters 12 and Clusters 29 recapitulate mitochondrial complexes: ATP synthase (red text) and the COX complex (green text). The nodes are colored by their lineage and the presence of a purple border indicates a predicted mitochondrial targeting signal by TargetP. Diamond and circle shaped nodes indicate essential and dispensable proteins, respectively. (B) IFA of 306650–3×HA expressing parasites reveals colocalization with the F1B ATPase mitochondrial marker and not the parasite plasma membrane marker SAG1. (C) IP of: i) TGME49_306650 utilizing a sonication based method with table columns identifying prey, spectral counts across two replicates and FDR, and ii) TGME49_306650 utilizing 1% NP-40.

(FDR = 0.24; **Fig 4C** and **S10 Table**). Given that electron transport complexes are embedded in the inner mitochondrial membrane, we also performed IP from lysates prepared utilizing 1% NP-40 to optimize membrane complexes. In these solubilisation conditions, we identified a protein complex that consisted of many of the subunits predicted by Maclean, *et al* [80]. They predicted 8 total subunits in Complex II. Here we identify 5 of these subunits when we precipitate TGME49_306650–3×HA (**Fig 4C** and **S10 Table**). Interestingly, while they identified the canonical succinate dehydrogenase subunit B (SDHB) in their study, we detect neither SDHB nor SDHA. Finally, we tried to knock-out TGME49_306650 by replacing its endogenous locus with HXGPRT to create the full knockout strains, but the gene deletion did not generate viable parasites for both cases, suggesting that TGME49_306650 might be important for parasite fitness and tachyzoite survival.

## Complex predictions identify novel invasion proteins

ToxoNet predicts 18 clusters representing putative invasion complexes, including 1, 10 and 7 that are predicted to localize to micronemes, rhoptries and dense granules, respectively (e.g. clusters 16, 38, 67, 75, 76, 78, 83 and 86 in **Fig 3A**). Filtering for proteins with validated localization data reveals a set of 38 previously uncharacterized proteins, predicted to be involved in host invasion (**Table 1**). Of these, 15 (39%) are predicted to carry a signal peptide, consistent with their trafficking through the secretory pathway, while 45% are specific to the Apicomplexa lineage. Nine of these proteins have been localized to invasion organelles in ToxoDB user comments, of which two were localized to the IMC. Furthermore, 53% (20/39) are

**Table 1. Putative Novel Invasion Factors.** Columns indicate Gene IDs (TGME49), their product description, the prediction (+) of a signal peptide (SP), localization information from ToxoDB, predicted lineage and cluster(s) in which it occurs.

| Gene ID (TGME49) | Description | SP | Localization | Lineage | Clusters |
|---|---|---|---|---|---|
| 200360 | hypothetical protein | + | Dense Granules | Toxoplasma | 16 |
| 202620 | hypothetical protein | + | | Toxoplasma | 23, 78 |
| 204080 | histidine acid phosphatase superfamily protein | + | | Eukaryotes | 58, 62, 88 |
| 204340 | hypothetical protein | + | | Toxoplasma | 23 |
| 208370 | myosin heavy chain, putative | + | | Eukaryotes | 16, 23 |
| 215980 | hypothetical protein | | | Toxoplasma | 26, 46 |
| 218780 | phosphoserine aminotransferase, putative | | | Eukaryotes | 82 |
| 219250 | acetyltransferase, GNAT family protein | | | Eukaryotes | 82 |
| 221200 | CW-type Zinc Finger protein | | | Eukaryotes | 58 |
| 221480 | hypothetical protein | | Microneme | Toxoplasma | 23 |
| 224460 | aminopeptidase n, putative | + | | Eukaryotes | 65 |
| 225870 | hypothetical protein | | | Eukaryotes | 62, 88 |
| 230160 | hypothetical protein | | IMC | Apicomplexa | 23 |
| 230940 | hypothetical protein | | | Eukaryotes | 62 |
| 231160 | hypothetical protein | | | Coccidia | 30 |
| 242820 | hypothetical protein | + | | Eukaryotes | 67 |
| 244250 | hypothetical protein | | | Eukaryotes | 76 |
| 244690 | hypothetical protein | | | Eukaryotes | 65 |
| 247520 | TgWIP | + | PV localization | Eukaryotes | 75 |
| 248740 | hypothetical protein | | IMC | Eukaryotes | 58 |
| 249990 | hypothetical protein | | | Eukaryotes | 16, 23 |
| 253430 | asparagine synthetase, putative | | | Eukaryotes | 65 |
| 257380 | hypothetical protein | + | | Coccidia | 38, 83 |
| 258080 | hypothetical protein | | | Eukaryotes | 38 |
| 261440 | ARM repeats containing protein | | rhoptry surface | Eukaryotes | 23 |
| 268760 | hypothetical protein | + | | Toxoplasma | 88 |
| 268790 | hypothetical protein | + | | Toxoplasma | 67 |
| 269950 | hypothetical protein | + | | Coccidia | 65, 83 |
| 271740 | hypothetical protein | | | Eukaryotes | 58, 62 |
| 280370 | hypothetical protein | | | Eukaryotes | 23 |
| 293360 | hypothetical protein | | | Coccidia | 67 |
| 297070 | hypothetical protein | + | rhoptry | Eukaryotes | 22, 69 |
| 305070 | hypothetical protein | | | Coccidia | 22 |
| 309600 | hypothetical protein | | | Coccidia | 16 |
| 309760 | hypothetical protein | + | | Toxoplasma | 23, 78 |
| 315210 | rhoptry protein, putative | | | Eukaryotes | 76 |
| 316250 | hypothetical protein | + | | Coccidia | 16, 23 |
| 323100 | hypothetical protein | | | Coccidia | 16, 23 |

predicted to localize to dense granules, rhoptries and micronemes in hyperLOPIT datasets. These clusters also predict interactions between well characterized invasion factors and those with little known functional information beyond their localization in discovery screens (e.g., ROP11 and GRA32).

Focusing on specific complexes, cluster 16 predicts a novel dense granule complex, with many proteins annotated as localized to the dense granule (e.g. GRA12, GRA32, GRA35, GRA44 and a MAF1 homolog), together with several novel invasion proteins. For example,

TGME49_200360 is a hypothetical protein that has been previously localized to the parasitophorous vacuole (PV) [82]. The interacting proteins, GRA35 and GRA44, are also present in the overlapping cluster 78, suggesting that they may represent core subunits of both complexes. Cluster 67 is a dense granule complex containing GRA37, MAG1 and three hypothetical proteins. MAG1 is a known component of the cyst wall during the bradyzoite stage; however, none of the other proteins identified in this cluster were present in a recent proteomic survey of cyst wall components [83]. The presence of MAG1 in this complex might therefore represent a novel tachyzoite-specific function for this protein. Cluster 75 highlights four proteins that form an isolated subnetwork (i.e. they do not interact with any other network proteins) and include the known rhoptry proteins ROP40, ROP11 and ROP47. This cluster has the highest average SNP score and contains the two network proteins with the highest SNP scores, ROP47 and TgWIP (TGME49_247520), a cytoplasmic modulator of dendritic cell migration [84] (**Fig 5A**). Additionally, this cluster has the third highest average expression correlation (Pearson correlation = 0.833; **S9 Table**), suggesting that its function is tightly regulated. Cluster 76 also predicts a novel rhoptry complex that contains two relatively uncharacterized rhoptry proteins, ROP14 and RON1, and two hypothetical proteins. Integration of SNP scores reveals clusters 67, 75 and 78 are among the top ten clusters with the highest average SNP score. The distribution of SNPs for proteins in these clusters across previously designated clades A-F [9] (and a group of 18 unclassified strains) is visualized within node pie charts (**Fig 5A**). These can be used to generate hypotheses regarding the conservation of these complexes. For example, the high proportion of SNPs present in clade E in the protein TgWIP in cluster 75 suggest that associations with this protein might not be conserved and contribute to variation in the pathogenic phenotype of this relatively small clade of four strains.

## Putative IMC complexes recapitulate known structural organization and predict novel IMC-related proteins

Clusters 48, 71, 74, 77 and 89 represent putative IMC complexes and predict five novel IMC proteins (**Table 2**). Two of these proteins have been described as localized to the IMC in ToxoDB user comments, with TGME49_254870 specifically localized to the apical complex. One protein is predicted to localize to the IMC in hyperLOPIT datasets. To examine whether the rigid structural hierarchy of distinct apical, central and basal compartments was recapitulated in the coelution datasets, matrices of the highest pairwise PCCNM or WCC in any experiment were computed for all pair-wise IMC proteins. The highest pair-wise scores were utilized to account for potentially missing data points. Hierarchical clustering of these matrices successfully reconstructs this spatial sub-compartmentalization with proteins from the basal/central and apical compartments segregated into distinct clusters (**Figs 5B and S8**). The notable exception is ILP1 which has been previously reported to localize to the central compartment [82]. These results support the network's ability to predict IMC interactions that are largely consistent with its structural organization. Cluster 48 contains ISP3, IMC19 and IMC22. The spatial proximity of IMC19 and ISP3 has already been confirmed by BioID [63] while all three IMC proteins have been localized to the central subcompartment of the IMC [63,85]. Cluster 71 also recapitulates known IMC architecture in that both PHIL1 and ISP1 are localized to the apical subcompartment [85,86]. Cluster 74 represents a novel complex that contains IMC17, a centrally localized protein, and two functionally uncharacterized proteins (IMC-localized TGME49_242070 [87] and TGME49_267430). Together these results demonstrate the efficacy of ToxoNet at recapitulating meaningful interactions in the IMC.

To assess the quality of these predictions we performed IPs on *T. gondii* PRU strains engineered to have 3×HA tags at the endogenous loci of IMC17 and IMC19. In utilizing the

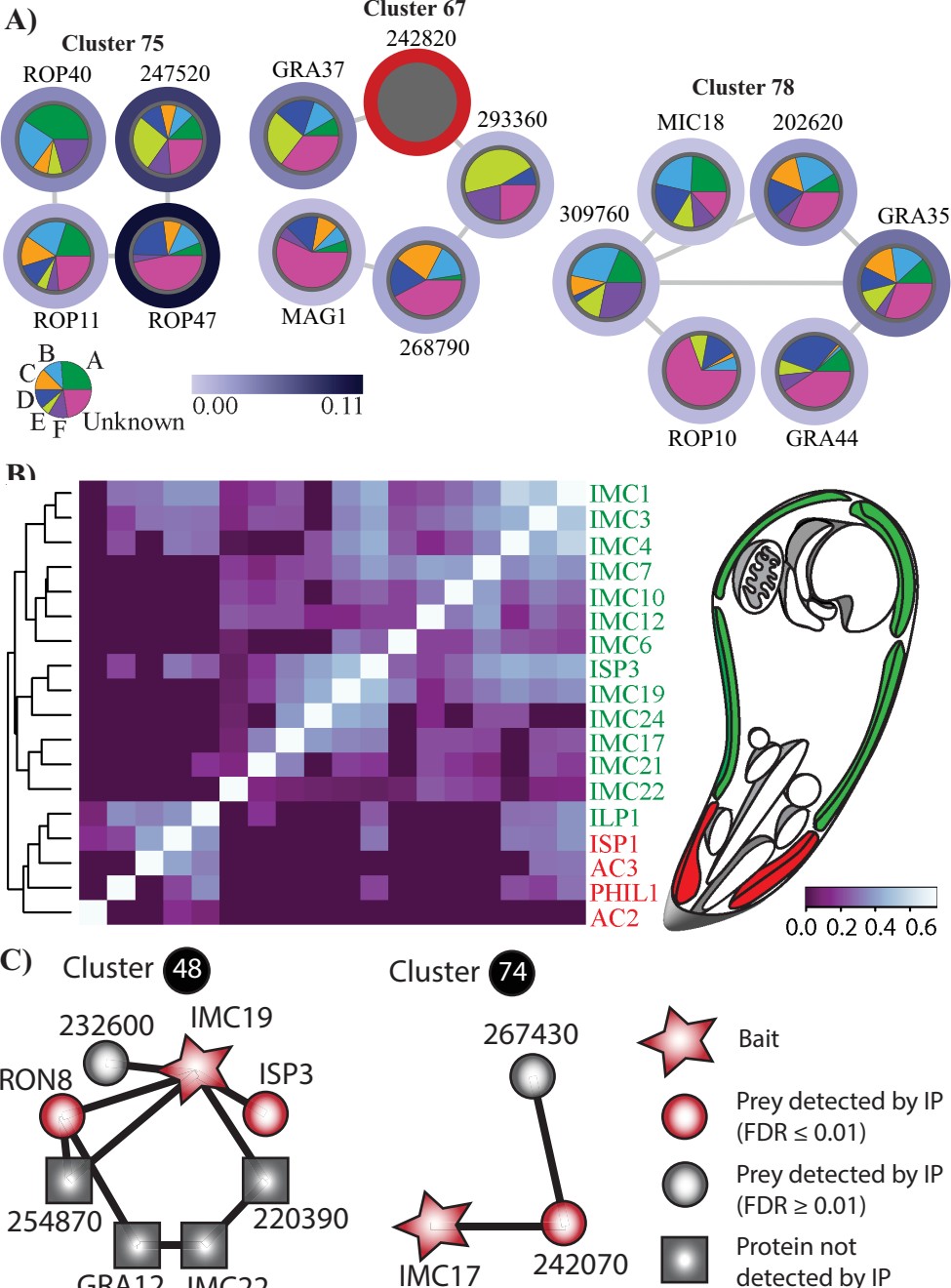

**Fig 5.** (A) These three putative invasion complexes are among the top ten clusters with the highest average SNP score. The proportion of SNPs present in each clade for each protein is indicated by the pie chart within each node. SNP scores for proteins, such as TGME49_242820, without reliable annotation in the ME49 reference genome were not calculated and are indicated with a red border. The legend indicating clade classification displays the proportion of 80 strains belonging to each of the 7 categories. (B) Heatmap showing coelution relationships between IMC proteins. For each protein pair, the highest PCCNM score from each coelution experiment for IMC proteins was used to generate a matric of coelution relationships. Clustering of these relationships recapitulates the sub-compartmental structural organization of the IMC. Only proteins with five or more spectral counts are considered. Proteins previously localized to the apical and basal/central subcompartments are highlighted in red and green, respectively. Hierarchal clustering was performed using the complete linkage method. (C) IP of IMC17, IMC19 and IMC22 with table columns containing identified prey, spectral counts and FDR.

**Table 2. Putative Novel IMC Proteins.** Columns indicate Gene IDs (TGME49), their product description, the prediction (+) of a signal peptide (SP), localization information from ToxoDB, predicted lineage and the cluster(s) in which it occurs.

| Gene ID (TGME49) | Description | SP | Localization | Lineage | Clusters |
|---|---|---|---|---|---|
| 220390 | hypothetical protein | | | Coccidia | 48 |
| 232600 | phospholipase, patatin family protein | + | | Apicomplexa | 48 |
| 254870 | hypothetical protein | | Apical Complex | Coccidia | 48 |
| 267430 | DnaJ domain-containing protein | | | Eukaryotes | 74 |
| 242070 | cAMP-dependent protein kinase regulatory subunit | | IMC | Eukaryotes | 74 |

sonication-based approach which generated ToxoNet we identified 230 and 190 significant preys (≤0.01 FDR), respectively. This large number of proteins in the IP eluent reflects the need for more targeted solubilisation methods in identifying meaningful IMC interactions; nonetheless, we were able to recapitulate network interactions (**Fig 5C** and **S10 Table**). Cluster 74 was particularly well recapitulated in the IP of IMC17 with TGME49_242070 significantly enriched and TGME49_267430 identified at an FDR of 0.17. IP was also performed on IMC19-3×HA transgenic strains to validate interactions in cluster 48. We detect ISP3 and RON8 significantly enriched in the eluent and TGME49_232600 at a higher FDR of 0.01. These results demonstrate the robustness of ToxoNet predictions and the ability to recapitulate our results with independent experimentation.

## Discussion

ToxoNet represents the first genome-scale protein-protein interaction network for *Toxoplasma gondii*. Based on coelution data for 1423 proteins across 6 experiments, we applied two machine learning strategies to integrate additional functional genomics datasets, including gene coexpression, phylogenetic profiles, and domain-domain interactions, to lend support to predicted interactions (**Fig 1**). The first strategy relies on a standard *supervised* approach, in this case Random Forest, which requires gold standard training datasets of known positive and negative interactions. Traditionally such training datasets have been challenging to generate for non-model organisms, typically relying on the inference of orthologous interactions that have been characterized for model organisms. Consequently, such datasets tend to be enriched for highly conserved proteins. In an attempt to overcome potential biases that such training data may generate, we therefore adopted a second strategy based on an *unsupervised* approach, in this case Similarity Network Fusion [41], which instead relies on a non-linear iterative strengthening or weakening of interactions based on their support across multiple complementary datasets. To our knowledge, this is the first instance of applying such an approach to predict protein-protein interactions. While we find that *supervised* approach, in general provides improved performance, we nonetheless find that the *unsupervised* approach performs significantly better than random (**S5 Fig**). Further, it predicted several previously characterized interactions that were not captured by the *supervised* approach. We therefore suggest that future protein-protein interaction studies also consider applying a similar two-pronged strategy. Further, such strategies might benefit from the evaluation of additional unsupervised approaches such as iCluster, MDI and others based on Bayesian approaches, kernel-based methods, and non-negative matrix factorization [88–90]. In addition, future studies may consider integrating complementary datasets such as hyperLOPT localization patterns [65]. Here we chose to use this dataset for validation purposes which precluded its use in network generation.

The resulting combined high confidence network consists of 2,063 high quality interactions between 652 proteins. While this number falls short of the 8,920 predicted proteins from the *T.*

*gondii* ME49 genome, previous screens of the tachyzoite stage have identified only 2,252 proteins as being expressed at this stage [91]. In this study, the sonication-based lysis conditions were optimized for soluble proteins. Proteomic screens of tachyzoite membrane proteins have identified 841 proteins [92]. The underrepresentation of membrane proteins has likely contributed to a relatively low number (1,423) of proteins identified in our fractions. In future studies, this technology can be adapted to optimize membrane complexes with non-ionic detergents or to investigate different lifecycle stages to increase coverage of the *T. gondii* proteome. Comparisons to a recent network of protein interactions generated for the related apicomplexan parasite, *Plasmodium* sp. [40] shared 22% of the interactions predicted by ToxoNet. This is surprisingly high considering that protein-protein interaction datasets rarely feature high levels of overlap, even in studies within the same organism using comparable methods [93]. Notable apicomplexan complexes predicted in both species, such as the Moving Junction and the Glideosome, are missing from the overlapping set of interactions and are better recapitulated in either the *T. gondii* or *Plasmodium* network, respectively. This suggests that technical differences, such as sample preparation and data processing, still limit cross-species comparisons of protein-protein interaction networks, that might otherwise identify instances of network 'rewiring' underlying species-specific adaptations to their specialized life cycles.

Application of the graph clustering algorithm, ClusterONE, predicts 93 clusters that recapitulate known, as well as novel protein complexes. Novel complexes successfully cluster proteins with known functional and spatial relationships, such as the organization of proteins that mediate pathogenesis, or localize to compartments, such as the mitochondria, cytosol and nucleus. It is noteworthy that one-third of the proteins identified by mass spectrometry are indicated as hypothetical by ToxoDB [94], and their identification in this coelution study provides evidence that they are expressed in the tachyzoite stage. Many of these hypothetical proteins and other poorly characterized proteins are present in protein complexes, representing valuable opportunities to drive new discoveries. Despite the limitations associated with generating a network on a non-model organism, the *T. gondii* network demonstrates robust congruence with parasite biology as demonstrated by benchmarking of additional datasets. This is evident with high correlation of transcript expression between pair-wise interactions, organization of essential proteins and agreement between spatial proximity of proteins in the cell and in the network.

Detailed analyses of identified protein complexes yield a number of novel insights into the organization of complexes with implications for both specialized parasite processes such as host invasion, as well as parasite-specific adaptations of otherwise highly conserved pathways. For example, many rhoptry and dense granule proteins are known to mediate important roles in maintaining the intracellular tachyzoite lifecycle, co-opting host machinery and regulating the host immune response. However, the full complement of these proteins remains to be elucidated and many that have been identified remain uncharacterized. Among these clusters are known invasion complexes, such as the MIC1/4/6 and GRA2/4/6 complexes, that additionally contain several novel components. These data also yield additional insights into the function of previously uncharacterized invasion factors, including GRA32, GRA37, GRA38, ROP11, ROP14, and ROP40, as well as predict many novel invasion proteins. Excitingly a recent study [95] cited our dataset in support of a novel tetrametric complex confirming our predictions of interactions between GRA32 (TGME49_212300), GRA70 (TGME49_249990) and GRA71 (TGME49_309600).

Outside invasion complexes, ToxoNet provides additional insights into the function and organization of the alveolate-specific IMC, an important organelle for host cell invasion and sexual reproduction [7]. Our coelution data reconstructs the rigid sub-cellular organization of

this compartment and as such, yields new testable hypotheses concerning the sub-cellular localization of novel IMC-related proteins and their complexes. Interestingly within our dataset we predict several interactions involving proteins from different compartments. Such findings are consistent with similar known examples, such as the well-established association between the microneme and rhoptry proteins: AMA1 and RON2 [68], and the interaction of other microneme proteins with the cytoplasmic protein, fructose-1,6-bisphosphate aldolase [69]. Such interactions in our dataset likely arise from the solubilization of compartments during sample preparation.

Beyond specialized pathways involved in invasion, several recent studies featuring fitness and proteomic screens, have identified an increasing number of apicomplexan-specific proteins that localize to the mitochondria, suggesting that this otherwise highly conserved organelle features a number of parasite-specific adaptations, particularly with respect to ATP synthase and oxidative phosphorylation [23,79,96,97]. This expanded view of the apicomplexan mitochondria is reflected in ToxoNet with the prediction of clusters that are supported by these previous studies. Here we identify and localize another novel apicomplexan-specific mitochondrial protein, TGME49_306650, that was previously uncharacterized. Its clustering with known ATP synthase and oxidative phosphorylation machinery suggests a role in these processes.

The construction of protein interaction networks provides a valuable scaffold onto which additional metadata may be integrated. For example, previous studies have leveraged protein interaction networks to inform on properties of essentiality and conservation, domain architectures, as well as taxon-specific representations of microbiome functionalities [34,98,99]. Here we show how ToxoNet can be used to interpret genetic variation information derived from 80 strains of *T. gondii*. Such approaches enable strain-level insights into the organization and function of protein complexes. In particular these visualizations allow us to distinguish proteins or protein complexes that represent conserved functions from those that underlie strain-specific functionality, which could be associated with host or tissue tropism.

## Conclusions

Here we present ToxoNet, the first high quality protein interaction network for *T. gondii*. We validate the quality of the network through systematic comparison of other protein interaction networks, both from other organisms as well as through the application of complementary technology. Our network predicts 93 clusters that capture well characterized complexes as well as complexes containing novel components and novel complexes. These data reveal a wealth of testable hypotheses and are provided here as a community resource.

## Materials and methods

### Culturing *T. gondii ME49* parasites and protein extract preparation

*T. gondii* ME49 parasites were cultured in two independent batches for 3 days in human foreskin fibroblast (HFF) cells supplemented by D10+M199 media. Parasites were harvested, washed with PBS and pelleted at 1500g at 4˚C for 15 min. The pellet re-suspended in 1 mL lysis buffer (10mM Hepes-pH 7.9, 1.5mM $MgCl_2$, 10mM KCl) with 1mM DTT and COmplete Mini Protease Inhibitor Cocktail (Roche). Lysates were subjected to sonication with 10 sec on, 10 sec off cycles at 30–35 W and centrifuged at 4˚C and 2000g for 20 min to clear cell debris.

### Pre-enrichment before HPLC Fractionation by affinity beads

For the material recovered from the second batch of cultured *T. gondii*, we used affinity beads (NuGel *PRO*spector) to pre-enrich *Toxoplasma gondii* lysate to capture five distinct sub-

proteomes. In this experiment, we added one volume of Cleanascite *PRO* to five volumes of the sample to remove lipids and any insoluble biomass. We then added PRO-BB binding buffer (pH 6.0) to the delipidated samples in a 1:1 volume ratio. The resulting mixture was then added to different reagent beads (*PRO*-A, *PRO*-C, *PRO*-L, *PRO*-N and *PRO*-R from NuGel *PRO*spector tool kit) in the Spin-X filterers (from NuGel *PRO*spector tool kit). The samples and beads were mixed for 10 min, and then centrifuged to collect the filtrate as 'flow-through' fractions. We then added PRO-BB binding buffer to wash the sample. We eluted the bound proteins by 200 ul elution buffer (0.2 M Tris, 0.5 M NaCl, pH 9.0). The buffer was exchanged for HPLC loading buffer by Zeba desalt spin column (Thermo) before HPLC fractionation. The resulting elutes from the five beads were kept for later HPLC fractionation.

## HPLC fractionation

We fractionated *T. gondii* cell lysate and enriched eluates from affinity beads using ion-exchange (IEX) liquid chromatography by an Agilent 1100 HPLC system (Agilent Technologies, ON, Canada) individually. A PolyCATWAX mixed-bed ion exchange column (200 x 4.6 mm id, 12 μm, 1500 A) was used with a 240 min salt gradient (0.15 to 1.5 M NaCl), for whole lysate samples prepared from the first batch of cultured *T. gondii*. For enriched eluate samples, a PolyCATWAX mixed-bed ion exchange column (200 x 4.6mm id, 5 μm, 1000A) was employed. Enriched samples were fractionated into 60 fractions by using a 120 min salt gradient (0.15 to 1.5 M NaCl). We collected the fractions every 2 mins. As a result, 120 fractions were collected for the whole cell lysate, and 60 fractions of each eluate from affinity bead. In total, 420 fractions were collected.

## LC-MS/MS analysis

All HPLC fractions were precipitated, re-dissolved and then digested by trypsin overnight at 37˚C resulting in peptides, which were subsequently dried and re-dissolved into 5% formic acid before LC-MS/MS. The LC-MS/MS was performed by a nano-flow HPLC System (EASY-nLC, Proxeon, Odense, Denmark) coupled with a LTQ Orbitrap Velos Mass Spectrometer (Thermo Fisher). First, the peptides were loaded into a 2.5 cm trap column (75 mm inner diameter), which was packed with Luna 5u C18, 100A beads (Phenomenex), by an auto-sampler. Next, a 10 cm analytical column (75 mm inner diameter), packed with 2 mm Zorbax 80XDB C18 reverse phase beads (Agilent), was connected to the trap column for peptides separation. A 60 min gradient of CAN in water (1% formic acid) from 5% to 35% was used to elute peptides. Electro-spray ionization was set at 2.5kV, and the mass spectrometer was operated in a data dependent mode (One full MS1 scan followed by MS2 acquisitions on top 10 precursor ions). The fragmentation was performed by 35% normalized collision energy at CID mode.

## Protein identification and label free quantification

We converted all raw files generated from LTQ Orbitrap Velos Mass Spectrometer to mzXML files by ReAdw software. The FASTA file was downloaded from ToxoDB v11 [100] and common contaminant peptides and corresponding reverse sequence decoys were added for false-discovery rate (FDR) evaluation. SEQUEST v2.7 [101] was used with default parameters to identify proteins with the probabilistic STATQUEST model subsequently applied to evaluate and assign confidence scores to all putative matches. Both proteins and peptides were considered positively identified, if detected within a 1% false discovery rate cut off (based on empirical target-decoy database search results). All the fractions were compared and concatenated using the Contrast software tool [102].

## Capturing similarity of coelution profiles

The similarity of coelution profiles for two proteins was estimated using various metrics. Three metrics implemented by Havugimana et.al were used as specified in [36]: a) pearson correlation coefficient with noise modeling (PCCNM)–which introduces random noise into the profiles in order to negate the high spurious correlations arising due to low abundance; b) weighted cross correlation (WCC)–which negates minor spectral shifts arising in the coelution profiles during collection of fractions; c) overall coapex score (Coapex1)–which captures the number of coelution experiments in which the same fraction contains peak abundance for both proteins. In addition, three other scores were implemented: a) mutual information (MI), implemented using the "entropy" package in R, in order to capture non-linear dependencies between the two coelution profiles; b) topological overlap similarity (TOM)–typically used in gene coexpression networks [103] to measure the relative interconnectedness between two proteins by combining the similarities between the two proteins (captured via their PCCNM, WCC scores) along with those of its shared neighbours–implemented using the scran and WGCNA packages in R; and c) individual coapex score (Coapex_X)–capturing the number of fractions with same peak abundance for both proteins in an experiment. In summary, for each protein pair, 6 PCCNM scores, 6 WCC scores, 6 Coapex_X scores, 6 MI scores, 6 TOM-PCCNM, and 6 TOM-WCC scores are generated, corresponding to each of the 6 experiments (five from bead purifications and one from whole cell lysate). Further, apart from Coapex1 (which is calculated over all experiments), overall scores were calculated using two approaches for the other measures by capturing similarity over all experiments (420 fractions) after ensuring that the two proteins coelute in at least one experiment: a) Calculate similarities over 420 fractions (PCCNM_overall, WCC_overall, MI_overall, TOM-PCCNM_overall, TOM-WCC_overall); and b) Calculate similarities only over the experiments in which either / both of the proteins are detected (PCCNM_overall-E, WCC_overall-E, MI_overall-E). Note that the latter was not carried out for TOM-based scores which considers shared neighbours in each experiment, resulting in too many combinations of experiments, with several being populated sparsely. Details of scores are shown in **S3 Table**.

## Integrating functional genomics datasets

The coelution datasets are integrated with additional functional genomics data for *T. gondii* using a supervised machine learning classifier in order to generate a protein interaction network. The additional datasets are:

a. Domain-domain interactions: Lee et. al. [46] generated log-likelihood scores to capture how often two Pfam domains [104] found in two proteins are involved in physical interaction. An integrated log-likelihood score was generated for each protein pair based on the log-likelihood scores assigned for their Pfam domain.

b. Phylogenetic profiles–Phylogenetic profiles represent the pattern of gene/protein distribution in sequenced genomes. Phylogenetic profiles for *T. gondii* were obtained from Phylo-pro v2 [105], which stores gene/protein distribution for 165 eukaryotic proteins [106]. Two types of phylogenetic profiles were generated: i) where presence/absence of orthologs is represented as 1s and 0s ii) where presence of homolog is represented in terms of modified BLAST E-value reflecting extent of sequence similarity, thereby capturing more information than binary gene presence/absence [45]. The similarity of phylogenetic profiles of two proteins was estimated using mutual information, implemented by the entropy package in R.

c. Coexpression–Coexpression of *T. gondii* proteins in 44 conditions was obtained from 8 coexpression datasets, from ToxoDB v11 [94]. The datasets were split into two based on the method: RNAseq and Microarray. The similarity between two coexpression profiles was estimated using pearson correlation coefficient. The datasets considered are: Buchholz_-Boothroyd_M4_in_vivo_bradyzoite_rnaSeq, Sibley_ME49_bradyzoite_rnaSeq, DBP_Hehl-Grigg_rnaSeq, Knoll_Laura_Pittman_rnaSeq, microarrayExpression_Boothroyd_Life-Cycle, microarrayExpression_Matrajt_GSE23174_Bz, microarrayExpression_Roos-Tz, microarrayExpression_Sullivan_GSE22100_GCN5-A.

d. Gene fusion and Textmining–Protein pairs in *T. gondii* reported to be functionally interacting according to gene fusion and text-mining (co-citation) evidence from STRING v10 were considered [107]. The scores representing their similarity in the STRING database were considered as is.

## Supervised machine learning approach to network generation

A supervised machine learning approach based on a RandomForest classifier (implemented using weka software suite v3.6) was used to integrate the similarity scores of coelution and functional genomic datasets using a training dataset of 511 positive interactions and 1533 negative interactions generated from ToxoCyc [108] and 1:1 orthologues of manually curated complexes from Cyc2008 (http://wodaklab.org/cyc2008/), CORUM [109] (**S4 Table**). The positive training dataset consists of 333 pairs from Toxocyc, 133 pairs sharing same GO-Biological Process terms (GO term annotation by Pfam2GO taken from [9]), 206 pairs from orthologs of yeast complexes in Cyc2008, and 516 pairs from orthologs of human complexes in CORUMCore (complexes with 50 members). Duplicate pairs and ribosomal pairs were removed from these datasets, yielding an overall positive training dataset of 511 pairs. The negative training dataset was generated by choosing pairs that do not share common localization (according to Apiloc v3 annotation wherever available (152 pairs), or by comparison of GO terms of orthologs in Cyc2008 and CORUM otherwise (2227 pairs and 1912 pairs respectively)). We initially evaluated several classifiers with the training dataset using 10-fold cross validation: RandomForest, SVM, BayesNet and found that RandomForest yielded the best performance in terms of AUC. Of the 52 features in total comprising both coelution (45 scores) and functional genomics (seven scores) similarity scores, feature selection identifies 33 attributes as providing information gain with respect to the training data (**S3 Table**). The RandomForest classifier was retrained based on these 33 features using 10-fold cross validation i.e. the training dataset was randomly split into 10 sets, with each of the 10 iterations using a different set as the test data, while the remaining 9 sets are used to generate the training data. The RandomForest classifier was trained using 100 trees, each constructed while considering five random features, at a maximum tree depth of 0 (corresponding to unlimited). The -I hyperparameter (number of trees in the RandomForest) was varied from 100 to 500, in steps of 100. However, since the AUC was not significantly impacted across iterations, we chose the number of trees (100) which gave the best performance in terms of overlap with previously known identified complexes. Averaging classifier results following the 10-fold cross validation gave an AUC score of 0.806 (**Fig 1D**). This RandomForest classifier was used to evaluate the set of 174455 protein pairs with biochemical support (coelution score $\geq 0.5$ in any of the experiments). The resulting set of predicted interacting pairs were further pruned to: i) remove pairs containing proteins with a spectral count of $\leq 5$ (to remove spurious correlations arising out of low abundance—90% of the scores with a value of–nan according to PCCNM have $\leq 5$ spectral counts in one of the protein pairs); and ii) remove pairs of proteins not containing any unique peptides distinguishing one from the other.

## Unsupervised machine learning approach to network generation

An unsupervised machine learning approach, based on Similarity Network Fusion (SNF) [41], was also adopted for integrating the coelution similarity scores with functional genomics similarity scores. Initially, using the set of 1329 proteins common to the coelution, coexpression and phylogenetic profile datasets, seven individual similarity networks were generated for each of the following scores: three similarity measures for coelution data (PCCNM_overall, WCC_overall, Coapex1), two similarity measures for coexpression data (COEXPR_RS, COEXPR_MA), and two similarity measures for phylogenetic profiles (MI based on presence/absence (MI-PresAbs) and sequence similarity metric (MI-pij)). We evaluated the contribution of the different datasets to network generation by SNF by examining different combinations of these seven similarity networks (**S4 Fig**). Note we examined the removal of COEXPR_MA and MI_PresAbs, due to overlaps with COEXPR_RS and MI_pij, respectively and perceived quality of the latter two datasets relative to the former. During generation of these networks, we systematically examined combinations of three hyperparameters used by SNF: 1) the number of neighbours, K (increased from 2–30 in increments of 2); 2) hyperparameter alpha (increased from 0.2 to 1 in increments of 0.2); and 3) number of fusion iterations, I (increased from 2–30 in increments of 2). The resulting set of integrated pairs were pruned at a false positive rate of 0.0001 (determined based on the training dataset generated for the supervised learning technique). Once generated, networks were clustered with the Markov Clustering Algorithm (MCL) [110] using an inflation parameter of 2.6. Resulting clusters were analysed for overlap with known protein complexes and pathways (orthologs of CORUM/Cyc2008 complexes and members of same pathway in ToxoCyc) using the Bader-Hogue scoring scheme [70]; clusters with a cutoff of $\geq$ 0.25 were considered to be similar [55]. From these analyses we found that the network generated from the combination of coelution and functional genomics scores, with the exception of MI_PresAbs, performed best (**S5 Fig**) and was therefore used as the *unsupervised* network in downstream analyses. Subsequent to network generation, we applied a similar pruning approach as for the supervised network: i) remove pairs containing proteins with a spectral count of $\leq$5; and ii) remove pairs where there are no unique peptides distinguishing one protein from the other. Finally, only protein pairs with a PCC $\geq$0.5 in COEXPR_RS were considered in order to generate the high confidence unsupervised network (**S6 Table**).

## Generating combined network and combined high-confidence network

The overlap similarity score [70] was used to identify the best unsupervised predicted networks in terms of their ability to identify biologically relevant clusters: From the various unsupervised predicted networks generated for different combinations of similarity matrices and parameters, the network with the highest number of clusters according to the overlap score cutoff ($\geq$0.25) was selected. For the supervised network, the network regenerated using the 33 features identified to be informative based on feature selection (see above) was chosen. These two networks were integrated to generate the overall combined network. Further, to generate a high confidence set, the supervised network was derived by considering interactions with RandomForest predicted scores greater than a cutoff corresponding to a false positive rate of ~0 (indicated by a * on the ROC curve–Fig 1). For the unsupervised network, the high confidence version was generated by considering all those interactions that are coexpressed (PCC $\geq$0.5 for COEXPR_RS). These two high confidence networks were collated in order to generate a combined high confidence network (**S7 Table**).

## Network analysis

Networks were visualized using Cytoscape v3.7.1 [111]. Topological analysis of the network was achieved using the Cytoscape plugin NetworkAnalyzer to determine node degree, node

betweenness, network degree distribution, network characteristic pathlength and network shortest pathlength distribution. Essentiality information was integrated from a genome-wide CRISPR screen of the tachyozoite lifecycle [23]. Essential proteins were defined by a phenotype score of less than or equal to -2 based on the reported separation of proteins with experimentally validated essential and dispensable fitness phenotypes [23]. The *T. gondii* network was compared to previously published *T. bruceii* [64] and *Plasmodium* [40] networks using 1:1 orthologs identified by InParanoid v4.1 [112]. Information from spatial proximity experiments was analyzed in the context of the network. In publications that utilized both BioID and APEX techniques [60,61], the subnetwork was defined by the network proteins present in the overlapping set detected by both methods. In cases where only BirA* fusion proteins were utilized, the subnetwork was defined by network proteins detected in streptavidin pull-downs that were not present in negative controls [62,63]. The Python package networkx was used to calculate the pairwise pathlength of all subnetwork proteins and the average was the characteristic pathlength of the subnetwork. The 55 network proteins that are isolated in clusters of 2–4 proteins were not included in this analysis due to their disconnection from the main network. Subnetwork permutations were generated by randomly selecting the same number of proteins from the network and determining the characteristic path length. This was repeated 1000 times for each subnetwork. P-values were determined using the pnorm function in R. For the G.O. comparisons, computed G.O. component, processes and function terms were taken from ToxoDB (downloaded April 5th 2022). Of the number of interactions where both proteins were annotated, the percent of interactions with the same term was calculated and significance of congruence of terms calculated using the Fishers exact test. To analyze coexpression correlation of network interactions, the following RNA-seq datasets were retrieved from ToxoDB: 'tachyozoite transcriptome time series (ME49)', 'tachyozoite transcriptome 3 and 4 days post-infection (VEG NcLIV)' [11] and 'comparative ribosome profiling of intracellular and extracellular parasites' [66]. The Pearson correlation for pair-wise transcripts was calculated for transcript levels in transcripts per million (TPM) values across 10 timepoints. TPM values were for reads that mapped back to the transcript sense strand, including non-unique reads. Pearson correlation values were determined for pair-wise network interactions, all pair-wise combinations of network proteins that were not predicted to be interacting, and 10,000 random pair-wise combinations from the full proteome. The significance of the difference in the distributions of these respective values was determined by a two-sample Kolmogorov-Smirnov test.

## Clustering and cluster analysis

Clusters were identified in the combined high confidence network using ClusterONE [55] using a density parameter of 0.25. The density parameter was optimized by iteratively scanning each possible value. To select a final cluster set, the number of recapitulated complexes utilized in supervised training data with an overlap score of 0.25 or greater [70] was calculated. The average number of recapitulated training complexes was also calculated from 100 random cluster permutations. The set of clusters that recapitulated the greatest number of training complexes minus the average number of training complexes recapitulated at random was selected as the final set of clusters. For the analysis of coexpression of cluster proteins, the average cluster coexpression correlation was calculated for each cluster by averaging the Pearson correlation for each pairwise combination of cluster proteins. The average correlation with network proteins was calculated by averaging the Pearson correlation for each pairwise combination of cluster proteins with all network proteins. The p-value was calculated using a Welch's T-test to determine the significance of the difference of the average coexpression correlation of cluster proteins with each other relative to all network proteins. For the analysis of distribution

of essential proteins in clusters, the fraction of essential proteins for each cluster was calculated. 100 random cluster permutations were generated as previously described. The significance of the difference in the distributions of essential proteins in clusters and random clusters was determined by a two-sample Kolmogorov-Smirnov test.

## SNP analysis

The dataset of SNPs used here compiles sequencing results from previous studies of *T. gondii* genetic diversity [9,73], and contains 1,054,454 SNPs spanning 80 different strains of *T. gondii*. The effect of these substitutions on the ME49 reference genome were assessed using SnpEff v4.3 [113], which produced multiple functional annotations (including the affected gene and type of mutation) for each given SNP. SNPs with minor alleles only present in 3 or fewer strains were excluded from further analyses. The SNP dataset was further filtered down by considering only those SNPs causing non-synonymous mutations for proteins in the network, resulting in a final set of 2893 SNPs. To determine which of these SNPs have pronounced biological effects, the 2881 network SNPs annotated as missense mutations were each assigned a score using PROVEAN v1.1.3. SNPs with a PROVEAN score of -2.5 or below were considered to be putatively deleterious. The SNP score was calculated for each network protein as the number of non-synonymous SNPs normalized by the protein length. The deleterious SNP was similarly calculated, but including only non-synonymous SNPs that were flagged as deleterious by PROVEAN. The previously described neighbor network [9] designates 62 of these *T. gondii* strains in clades A-F; the remaining 18 are designated as unknown. For each of the non-synonymous SNPs, of the strains carrying the minor allele, the proportion belonging to each clade was calculated. The proportions for each clade were averaged for all these SNPs within a protein to determine the final clade distribution visualized within node pie charts. The average SNP score of each cluster was determined by averaging the SNP score of each cluster protein.

## IMC reconstruction

To relate cofractionation datasets to characterized structural hierarchies of the apicomplexan IMC, PCCNM or WCC matrices were computed for all known IMC proteins contained in the proteomic data with five or more spectral counts. These matrices contained the highest pairwise PCCNM or WCC score from individual fractionation experiments (i.e., five affinity beads and IEX) or from across all 420 fractions, respectively. Hierarchical clustering of these matrices was performed using the complete linkage method and heatmaps and representative dendograms were generated using the heatmap.2 function in R.

## Endogenous epitope tagging, immunofluorescence and immunoprecipitation

*T. gondii* Pru Δku80/Δhxgprt strain was used to generate the endogenous 3×HA-tagged strains used for IFA assays. The 3×HA sequence was inserted into the endogenous locus of TGME49_306650 by double crossover homologous recombination using CRISPR/Cas-based genome editing and homology arms of 40 bp to facilitate the 3×HA integration. For all transfections, 6 μg of guide RNA was transfected along with 3 μg of a repair oligo (3xHA). Parasites were transfected and cloned by limiting dilution after the first lysis. Insertion of 3xHA tag was confirmed by conventional PCR and IFA. All strains were maintained in HFF cells grown in Dulbecco's modified Eagle's medium (DMEM) supplemented with 10% heat-inactivated fetal bovine serum (FBS), 0.25 mM gentamicin, 10 U/mL penicillin, and 10 μg/mL streptomycin (Gibco, Thermo Fisher Scientific Inc., Grand Island, NY). For IFA, HFF monolayers were infected with *T. gondii* for 24 hours. Coverslips were then fixed with 0.1% Triton X-100 and

incubated with primary antibodies. The signal from primary antibodies was then amplified using species-specific secondary antibodies conjugated to Alexa488, Alexa647 and Rhodamine Red X. DNA was stained using DAPI. Imaging was performed utilizing an Olympus IX81 quorum spinning disk confocal microscope with a Hamamatsu C9100-13 EM-CCD camera (the Imaging Facility at the Hospital for Sick Children). Lysates for IP were prepared as described above or by incubating *T. gondii* pellets in lysis buffer (1% NP-40, 150 mM NaCl, 50 mM Tris pH 8.0, 5 mM EDTA) for 1 hour at 4˚C and cell debris was removed by centrifugation. In addition to IP of strains with endogenously tagged proteins, parallel IPs were performed on *T. gondii* PRU WT for negative controls. The IP was performed utilizing SureBeads Protein G magnetic beads (BioRad); each with a total of 2 replicates. The resulting quantitative mass spectrometry data was analyzed utilizing SAINTexpress to identify high confident hits (FDR $\leq$ 0.01).

## Supporting information

**S1 Fig. Precision-Recall curve for the Randomforest approach, for different combinations of features.**
(PDF)

**S2 Fig. Predicted high confidence supervised network depicted using cytoscape, where proteins are shown as nodes (circles) and the predicted interactions between them are shown as edges (lines).** Hypothetical proteins are indicated as squares enclosed by black borders. Well known protein complexes are colored uniquely and encircled–such as ribosome (yellow), proteasome (orange), snRNP complex (light green), glycolytic complex (blue), prefoldin complex (black). Proteins known to be involved in invasion are colored pink, and proteins associated with the IMC are colored purple.
(PDF)

**S3 Fig. Performance of networks generated from the integration of different combinations of datasets by SNF.** After network generation, clusters were defined using the MCL algorithm and evaluated for overlap with known protein complexes using the Bader-Hogue overlap scoring algorithm. The graph shows the number of unique clusters with overlap score $\geq$0.25 with respect to known protein complexes for six combinations of networks: 1) coelution scores (PCCNM, WCC and Coapex1); 2) phylogenetic scores (MI_PresAbs, MI_pij); 3) functional genomics scores (MI_PresAbs, MI_pij, COEXPR_RS and COEXPR_MA); 4) coelution and functional genomics scores (PCCNM, WCC, Coapex1, MI_PresAbs, MI_pij, COEXPR_RS and COEXPR_MA); 5) coelution and functional genomics scores excluding COEXPR_MA and MI_PresAbs; and 6) coelution and functional genomics scores excluding MI_PresAbs. In this analysis, phylogenetic profile scores performed better than coelution scores. However, overall we found the network generated from combining coelution and functional genomics scores, excluding MI_PresAbs, gave the best performance. This network was selected as the final *unsupervised* network.
(PDF)

**S4 Fig. Fraction of interactions lost for various cutoffs of coexpression measure (PCC) for both the supervised and unsupervised networks.**
(PDF)

**S5 Fig. Comparison of supervised and unsupervised networks.** (A). Venn diagram showing the number of overlapping and unique proteins for the supervised and unsupervised networks. (B) Features of proteins uniquely identified by the unsupervised network: Distribution of

spectral counts for the proteins, Box plot comparing the PCCNM scores for the interactions of these proteins with an equivalent set of randomly generated interactions (C). Box plots depicting the distribution of various coelution, coexpression, and phylogenetic scores for Supervised, Unsupervised, and an equivalent set of Randomly generated interactions. The box and whiskers in each boxplot indicate the 25%-75% quartile and min-max of the scores over all the interactions in a dataset, respectively.
(PDF)

**S6 Fig.** (A) The distribution of random permutations (n = 1000) of characteristic path lengths relative to the actual characteristic path length (red line) of network proteins identified in biotinylation BirA-based BioID experiments with GRA13 (n = 35, p = 0.43), GRA17 (n = 48, p = 0.1), GRA25 (n = 71, p = 0.07), ISP3 (n = 46, p = 0.85), AC2 (n = 14, p = 0.21). (B) The intersection of interactions predicted in recent *Trypanosome bruceii*[6] and *Plasmodium falciparum*[7] protein interaction networks.
(PDF)

**S7 Fig. Characteristics of essential and dispensable proteins.** (a) Overlapping histograms compare the distribution of the fraction of conserved eukaryotic proteins in essential and dispensable clusters. Boxplots illustrate the fraction of invasion and IMC proteins (b) and hypothetical proteins (c) in essential and dispensable clusters.
(PDF)

**S8 Fig.** (A) Distribution of the average SNP score for each predicted protein complex. (B) Hierarchal clustering of the best pairwise WCC score from each coleution experiment for IMC proteins recapitulates the subcompartmental structural organization of the IMC. Proteins previously localized to the apical and basal/central subcompartments are highlighted in red and green, respectively.
(PDF)

**S1 Table. Details of peptide counts obtained from mass spectrometry analysis of *T. gondii* tachyzoites from six cofractionation experiments.**
(XLSX)

**S2 Table. Details of *T. gondii ME49* proteins identified in the cofractionation experiments, along with their elution profiles.**
(XLSX)

**S3 Table. Description of scores evaluated in the study along with selected scores from feature selection.**
(XLSX)

**S4 Table. Description of POSITIVE and NEGATIVE training datasets used for training the supervised classifier.**
(XLSX)

**S5 Table. Coelution, coexpression, and phylogenetic profiles of predicted interactions for the supervised network.**
(XLSX)

**S6 Table. Coelution, coexpression, and phylogenetic profiles of predicted interactions for the unsupervised network.**
(XLSX)

**S7 Table. List of interactions comprising the combined high confidence network.**
(XLSX)

**S8 Table. Summary of correspondence in hyperLOPIT predictions for interacting pairs of proteins in the combined high confidence network.**
(XLSX)

**S9 Table. Details of Protein Complexes including, protein components, complex size, name of recapitulated training complex, overlap score of training complex, fraction of eukaryotic, apicomplexan, coccidian and toxoplasma-specific proteins, fraction of essential and dispensable proteins, putative localization, average SNP score, average coexpression correlation of cluster proteins, the average coexpression correlation of cluster proteins with all network proteins and the p-value determining the significance complex coexpression.**
(XLSX)

**S10 Table. Affinity-purification mass spectrometry (AP-MS) results processed using SAINTexpress for IPs of TGME49_306650, IMC17 and IMC19 baits.** The 'Bait', 'Prey' and 'Prey Gene' columns indicate the bait, identified prey and the identified prey's gene name, respectively. The 'ToxoNet HC' and 'ToxoNet LC' columns indicate whether the prey is a predicted interactor of the bait in the ToxoNet high confidence and low confidence networks, respectively. The 'ToxoNet Cluster' column indicates the ToxoNet cluster that contains both the prey identified by AP-MS and the bait. The 'Spectral Counts', 'Number Replicates', 'Control Spectral Counts' and 'BFDR' columns indicate the number of spectral counts for each prey identified in the bait eluent, the number of biological replicates for the bait, the number of spectral counts for each prey identified in the negative control eluent and the Bayesian false discovery rate, as determined by SAINTexpress, respectively.
(XLSX)

## Acknowledgments

Computing resources were provided by the SciNet HPC Consortium. SciNet is funded by: the Canada Foundation for Innovation under the auspices of Compute Canada; the Government of Ontario; Ontario Research Fund—Research Excellence; and the University of Toronto. We would like to thank Peter Bradley (UCLA) for generously providing us with α-F1B ATPase antibody and two transgenic *T. gondii* PRU strains (IMC17-3×HA and IMC19-3×HA).

## Author Contributions

**Conceptualization:** Michael E. Grigg, Andrew Emili, John Parkinson.

**Data curation:** Lakshmipuram S. Swapna, Grant C. Stevens, Lucas Zhongming Hu, Daniel D. Fusca, Xuejian Xiong.

**Formal analysis:** Lakshmipuram S. Swapna, Grant C. Stevens, Aline Sardinha-Silva, Lucas Zhongming Hu, Daniel D. Fusca, Xuejian Xiong.

**Funding acquisition:** Jon P. Boyle, Michael E. Grigg, Andrew Emili, John Parkinson.

**Investigation:** Lakshmipuram S. Swapna, Grant C. Stevens, Aline Sardinha-Silva, Lucas Zhongming Hu, Xuejian Xiong.

**Methodology:** Lakshmipuram S. Swapna, Grant C. Stevens, Aline Sardinha-Silva, Lucas Zhongming Hu, Verena Brand, Cuihong Wan, Xuejian Xiong, Jon P. Boyle.

**Project administration:** Michael E. Grigg, Andrew Emili, John Parkinson.

**Resources:** Lucas Zhongming Hu, Verena Brand, Cuihong Wan, Jon P. Boyle, Michael E. Grigg, Andrew Emili, John Parkinson.

**Software:** Lakshmipuram S. Swapna.

**Supervision:** Cuihong Wan, Jon P. Boyle, Michael E. Grigg, Andrew Emili, John Parkinson.

**Validation:** Grant C. Stevens, Aline Sardinha-Silva.

**Visualization:** Grant C. Stevens.

**Writing – original draft:** Lakshmipuram S. Swapna, Grant C. Stevens, John Parkinson.

**Writing – review & editing:** Lakshmipuram S. Swapna, Grant C. Stevens, Aline Sardinha-Silva, Lucas Zhongming Hu, Verena Brand, Daniel D. Fusca, Michael E. Grigg, Andrew Emili, John Parkinson.

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
