## [Decision Letter · Decision Letter 0]

29 Feb 2024

Dear Dr. Parkinson,

Thank you very much for submitting your manuscript "ToxoNet: A high confidence map of protein-protein interactions in Toxoplasma gondii" for consideration at PLOS Computational Biology.

As with all papers reviewed by the journal, your manuscript was reviewed by members of the editorial board and by several independent reviewers. In light of the reviews (below this email), we would like to invite the resubmission of a significantly-revised version that takes into account the reviewers' comments.

All three reviewers raised similar issues but agree that the work is of potential interest to the field. Please respond productively to all reviewers' comments. Please pay close attention to comments regarding the issue of the apparent low correlation between hyperLOPIT predicted localization in identified complexes (i.e. many complexes have components that are apparently in different compartments).

We cannot make any decision about publication until we have seen the revised manuscript and your response to the reviewers' comments. Your revised manuscript is also likely to be sent to reviewers for further evaluation.

Sincerely,

Michael L Reese, PhD

Guest Editor

PLOS Computational Biology

Stacey Finley

Section Editor

PLOS Computational Biology

All three reviewers raised similar issues but agree that the work is of potential interest to the field. Please respond productively to all reviewers' comments. Please pay close attention to comments regarding the issue of the apparent low correlation between hyperLOPIT predicted localization in identified complexes (i.e. many complexes have components that are apparently in different compartments).

Reviewer's Responses to Questions

**Comments to the Authors:**

Reviewer #1: Toxonet is an excellent resource for the toxoplasma community, combined with some interesting analysis I think this is a great paper. It includes new data, new appproaches and follow up analysis in several cases.

I have some minor comments that I hope that authors can address.

1) Could the authors, where possible, show the pointers underlying the data - i.e. when they present error bars?

2) The authors show results improve with adding more data quite convincingly but how SNF compares with other approaches for data integration is not clearly presented in the manuscript.

3) 38% percentage of interacting proteins being in the same compartment appears quite low - is there any explaination for this? Did the authors think about integrating this data also?

4) It is unclear whether the experimental mass-spectrometry was replicated - as far a I understand the 480 fraction were generated once but the replicate to replicate variation wasn't examined?

5) The FDR of ~0 doesnt really quite make sense - do the authors really believe there are no false positives with there >2,000 interactions. I think some naunce is need here too, since proteins could co-elute but not necessarily form an interaction (and visa-versa if they come apart during lysis)

Reviewer #2: review is uploaded as an attachment (240210_Review.doc)

Reviewer #3: The manuscript from Swapna et al seeks to address the knowledge gap of protein-protein interactions (PPIs) and complex formation within Toxoplasma gondii. The manuscript is generally well written, with significant amount of experimental and computational work carried out to refine and validate the interactions identified. There are a few issues that need to be addressed before publication:

Major:

1. Page 7: The study reports 1423 unique proteins identified (supp table 2), however, over 100 proteins have been identified with only a single spectrum across 420 fractions. These are not reliable identifications and should be removed from the results.

2. Page 8: Gold standard negative interactions were generated based on differences in cellular localization. However, in page 15 and table 1, only 38% of interacting pairs shared the same localization according to hyperLOPIT. Were the rest of 62% of interacting pairs based in different localization or not identified in hyperLOPIT dataset? hyperLOPIT information should be added when generating negative controls.

3. Page 14: “Pearson correlation values were calculated for protein pairs across eleven time-points encompassing different lifecycle stages”. Since tachyzoite was the only lifecycle stages investigated in the study, adding expression data from other lifecycle stages would add noise to the analysis and protein interactions may not sustain across lifecycle stages, such as invasion related ones.

Minor:

1. Page 9 and figure 1C: “the ribosome, and a snRNP complex, demonstrating its ability to capture features in the training datasets (Figure 1C…” figure 1C appear to be a wrong reference to the sentence.

2. Page 12 and figure 2D: Essential proteins are highlighted as mediating more central roles in the network; however, the associated Figure 2D shows a lower betweenness centrality score for essential proteins.

3. Page 28: The 93 clusters identified should not be referred to as complexes. Comparing to experimentally validated complexes (page 16), there are many additional proteins grouped into the same cluster.

4. Page 28: Please provide more details on how the data was made freely available on ToxoDB, which was not easily searchable.

5. Page 31: Please provide parameters used in SEQUEST searching.

**Have the authors made all data and (if applicable) computational code underlying the findings in their manuscript fully available?**

Reviewer #1: Yes

Reviewer #2: None

Reviewer #3: Yes

PLOS authors have the option to publish the peer review history of their article (what does this mean?). If published, this will include your full peer review and any attached files.

Reviewer #1: **Yes: **Oliver Crook

Reviewer #2: No

Reviewer #3: No
---

## [Decision Letter · Decision Letter 1]

28 May 2024

Dear Dr. Parkinson,

We are pleased to inform you that your manuscript 'ToxoNet: A high confidence map of protein-protein interactions in Toxoplasma gondii' has been provisionally accepted for publication in PLOS Computational Biology.

Best regards,

Michael L Reese, PhD

Guest Editor

PLOS Computational Biology

Stacey Finley

Section Editor

PLOS Computational Biology

Nice work on the revision. Everything looks good to me.

Reviewer's Responses to Questions

**Comments to the Authors:**

Reviewer #1: The authors have answered my questions and I appreciate they're remarks on hyperLOPIT data. I'm convinced that the data and analysis will be extremely useful for the toxo community.

Reviewer #2: Thank you for addressing my comments, the revised manuscript is very nice.

Reviewer #3: This revision has addressed my previous comments. I recommend it to be accepted for publication.

**Have the authors made all data and (if applicable) computational code underlying the findings in their manuscript fully available?**

Reviewer #1: Yes

Reviewer #2: Yes

Reviewer #3: Yes

PLOS authors have the option to publish the peer review history of their article (what does this mean?). If published, this will include your full peer review and any attached files.

Reviewer #1: **Yes: **Oliver M. Crook

Reviewer #2: No

Reviewer #3: No

---

## [Editor Report · Acceptance letter]

12 Jun 2024

PCOMPBIOL-D-23-01674R1 

ToxoNet: A high confidence map of protein-protein interactions in Toxoplasma gondii

Dear Dr Parkinson,

I am pleased to inform you that your manuscript has been formally accepted for publication in PLOS Computational Biology. Your manuscript is now with our production department and you will be notified of the publication date in due course.

With kind regards,

Anita Estes
